

# Development of a Peltier-based chilled-mirror hygrometer for tropospheric and lower stratospheric water vapor measurements

Takuji Sugidachi[1], Masatomo Fujiwara[2], Kensaku Shimizu[1], Shin-Ya Ogino[3], Junko Suzuki[3] and Ruud J. Dirksen[4]

[1]Meisei Electric Co., Ltd., 2223 Naganumamachi, Isesaki-shi, Gunma, 372-8585, Japan
[2]Faculty of Environmental Earth Science, Hokkaido University, Kita 10 Nishi 5, Kita-ku, Sapporo, Hokkaido, 060-0810, Japan
[3]Japan Agency for Marine-Earth Science and Technology, Yokohama, Kanagawa, 236-0001, Japan
[4]GRUAN Lead Centre, Meteorologisches Observatorium Lindenberg, Deutscher Wetterdienst, Am Observatorium 12, 15848
Tauche, Germany

*Correspondence to*: T. Sugidachi (sugidachit@meisei.co.jp)

**Abstract.** We have developed a Peltier-based non-cryogenic chilled-mirror hygrometer named "SKYDEW" to measure water vapor from the surface to the stratosphere. Several chamber experiments were conducted to investigate the

characteristics and performance of the instrument under various conditions. The stability of the feedback controller that maintains the condensate on the mirror depends on the controller setting, the condensate condition, and the frost point in ambient air. The results of condensate observation by a microscope and proportional-integral-derivative (PID) tuning in a chamber were used to determine the PID parameters of the controller such that slight oscillations of the scattered light signal from the mirror and mirror temperature are retained. This allows for the detection of steep gradients in the humidity profile,

which are otherwise not detected because of the slower response. The oscillation of the raw mirror temperature is smoothed with a golden point method that select the equilibrium point of the frost layer. We further describe the details of the data processing and the uncertainty estimation for SKYDEW measurements in terms of the Global Climate Observing System (GCOS) Reference Upper-Air Network (GRUAN) requirements. The calibration uncertainty of the mirror temperature measurement is <0.1 K for the entire temperature range from –95 to 40 °C. The total measurement uncertainty of SKYDEW

measurements can exceed 0.5 K in the region where large oscillations of the mirror temperature remain.

Intercomparisons with relative humidity (RH) sensors on radiosondes, the cryogenic frost point hygrometer (CFH), and satellite Aura Microwave Limb Sounder (MLS) were performed at various latitudes in the Northern Hemisphere to evaluate the performance of SKYDEW. These results show that SKYDEW can reliably measure atmospheric water vapor up to 25 km altitude. Data from several SKYDEW and CFH measurements predominantly agree within their respective uncertainties,

although a systematic difference of ~0.5 K between SKYDEW and CFH was found in the stratosphere, the reason for which is unknown. SKYDEW shows good agreement with Aura MLS for profiles that are not affected by contamination.



# 1 Introduction

Atmospheric water vapor plays a critical role in the climate system because it acts as a medium for heat exchange and transport, and because it is linked to the formation of clouds and precipitation. It is also the dominant greenhouse gas, accounting for about 60% of the natural greenhouse effect under clear sky conditions and providing the largest positive feedback in model projections of climate change (Held and Soden, 2000). Stratospheric water vapor also makes a significant contribution to the radiative equilibrium of the stratosphere and small changes in stratospheric water vapor can have a large impact on Earth's radiation budget (Solomon et al., 2010). Thus, it is important to monitor and understand long-term variability in atmospheric water vapor in the troposphere and stratosphere. The distribution of water vapor in the atmosphere is highly variable, ranging from close to saturation to almost devoid of water, with complicated interconnected processes governing its actual distribution. Tropospheric water vapor concentration depends strongly on air temperature, and therefore on location and altitude. Usually near-surface air contains a high concentration of water vapor (typically >10000 ppmv (parts per million by volume) in the tropics), and the water vapor content typically reduces with height in the troposphere. In contrast, the stratosphere is extremely dry (typically <5 ppmv water vapor in the lower stratosphere) because tropospheric air enters the stratosphere typically through the cold (approximately –85 °C) tropical tropopause, causing freeze drying (Brewer, 1949). Brewer-Dobson circulation transports stratospheric gases and trace atmospheric components from the tropics to the poles, directly affecting the distribution of stratospheric water vapor. In addition to the dynamics process, there is also a chemical contribution to stratospheric water vapor changes, mostly by oxidation of methane (Myhre et al., 2007).

There are many methods for measuring atmospheric water vapor. Operational radiosondes employ a capacitive RH sensor. Because radiosondes are designed primarily to measure the tropospheric water vapor for the purpose of weather forecasting, the RH sensors on radiosondes perform poorly in the stratosphere. In addition, some operational radiosondes may have bias errors even in the troposphere due to time-lag errors, solar radiation effects in daytime and contamination errors under rainy conditions (Ingleby, 2017). Spaceborne observations provide information on the global distribution of water vapor, and thus represent an important source of information over the oceans, where radiosonde observations are scarce. The Microwave Limb Sounder (MLS) on the Aura satellite can observe atmospheric water vapor profiles above ~350 hPa (Read et al., 2022). Since the 1990s, the estimation of precipitable water vapor (PWV) from the atmospheric delay in the global navigation satellite systems (GNSS) signal has been available and is being continuously evaluated. The GNSS PWV retrieval has important advantages as an absolute measurement because it does not need independent calibration and is not affected by clouds (Shoji, 2013). However, this method estimates only the column integrated water vapor amount in the atmosphere, and thus cannot retrieve upper-air water vapor.

Raman lidars can retrieve high-resolution profiles of water vapor mixing ratio. Although the measurement range depends on aspects of the system configuration such as transmit power, the profiles have reached 14 km for 1-h integration times and 1.5 km vertical resolution, and can reach 21 km for 6-h integration times using degraded vertical resolution (Leblanc et al., 2012). The measurement accuracy is limited by calibration uncertainties (systematic errors) and by photon-



counting noise (random errors that increase rapidly with altitude). Raman lidar calibration can be performed by comparison with other collocated sensors, such as high-quality radiosondes (Whiteman et al., 2006). The calibration and validation of these remote sensing techniques requires high-accuracy "in situ" instruments.

The chilled-mirror hygrometer is a high-precision sensor employing thermodynamic principles that can measure water vapor between the surface and the stratosphere with high vertical resolution that cannot be achieved with remote sensing

technology. Consequently, the chilled-mirror hygrometer is also suitable for calibrating Raman lidar instruments. A chilled-mirror hygrometer directly measures the dew point or frost point temperature of the ambient air. The hygrometer actively maintains the equilibrium of two phases consisting of liquid water (or ice) on the mirror and water vapor in the adjacent air. The mirror temperature (i.e., the temperature of the condensate) is continuously adjusted so that the amount of condensate on the mirror is constant. When the condensate does not grow or shrink, the mirror temperature is expected to be equal to the

dew point or frost point temperature of the ambient air, depending on the phase of the condensate. The Clausius–Clapeyron equation (e.g., Murphy and Koop, 2005) is then used to convert the dew/frostpoint temperature to water vapor pressure. Most chilled-mirror hygrometers use an optical detection system to monitor the condensate amount on the mirror. Since the intensity of reflected/scattered light corresponds to the condensate amount, chilled-mirror hygrometers keep the intensity constant using a feedback controller (e.g., by using a proportional-integral-derivative (PID) controller), and measure the

mirror temperature continuously as the dew/frost point temperature. There are two main methods to control the mirror temperature: a combination of cooling by a cryogen and heating by a heater wire to achieve the desired mirror temperature (e.g., the CFH and the NOAA FPH), and cooling by a Peltier device, which can transfer heat from one side of the device to the other with consumption of electrical energy (e.g., the Snow White). A heatsink is attached to the hot side so that its temperature becomes close to the ambient air temperature by heat transfer. In this way, the mirror, which is attached to the

cold side, can be cooler than the ambient air and thus can reach the dew/frost point temperature of the ambient air.

Currently, two types of chilled-mirror instruments are used to routinely perform balloon-borne water vapour observations in the stratosphere. The Frost Point Hygrometer (FPH) developed by NOAA's Earth System Research Laboratory (NOAA/ESRL), which has been in use since 1980 at Boulder, USA (Mastenbrook and Oltmans, 1983; Hall et al., 2016; Hurst et al., 2022). This instrument provides the longest record of stratospheric water vapor (Hurst et al., 2011).

Although several improvements have been made to this instrument since 1980 (e.g., Vömel et al., 1995; Hall et al., 2016), the basic principles of the instrument have not changed, and it is believed that the data quality is almost constant. The Cryogenic Frostpoint Hygrometer (CFH) was developed in the mid-2000s, based on the NOAA FPH, and incorporated several improvements to enable it to measure water vapour concentrations in the troposphere and the stratosphere (Vömel et al., 2007a). Similar improvements were also adopted for the FPH. The CFH is currently available for purchase from

Environmental Science Corporation (EN-SCI). The NOAA FPH is not commercially available and is employed at a limited number of sites under a cooperation- agreement.

The CFH performance has been evaluated by comparison with various hygrometers, including NOAA FPH, the Aura MLS, Fluorescent Advanced Stratospheric Hygrometer for Balloon (FLASH-B) Lyman-alpha instrument, and Vaisala RS92



radiosonde (Miloshevich et al., 2006; Vömel et al., 2007a, 2007b, 2007c). The operating principle of both CFH and FPH is

that the mirror is connected to a cryogenic liquid (Trifluoromethane, $CHF_3$, commonly referred to as R23), providing conductive cooling and the mirror temperature is controlled by heating against this cold sink, using a heating wire at the back of the mirror, where the heating power is regulated by a PID controller. The boiling point of the cryogen determines the lower limit of the frostpoint that can be measured, which in case of R23 is -80 °C at sea level, and around -100 °C in the stratosphere. Because the production and use of $CHF_3$ was restricted by the Kigali Amendments of the Montreal Protocol in

2016, an alternative cooling method is required for continued use of the NOAA FPH and the CFH (Dirksen, 2020).

Such an alternate cooling method is employed by the Snow White instrument, a Peltier-based chilled-mirror hygrometer that has been manufactured by a Swiss meteorological instrument company, Meteolabor AG, since 1996 (Fujiwara et al., 2003; Vömel et al., 2003). The Snow White has a 3 mm × 3 mm mirror, which is at the same time a thermocouple thermometer, attached on the cold side of a Peltier device. The PID controller used to maintain the dew/frost layer is

implemented with an analogue rather than digital circuit. The uncertainty of the mirror temperature measurement is less than 0.1 K. The response time, which is largely determined by the time constant for vapor–water or ice–vapor equilibrium, ranges from negligible at +20 °C to 80 s at –60 °C (Fujiwara et al., 2003). The maximum temperature difference that can be achieved with Peltier cooling is temperature dependent, and in case of Snow White the largest dewpoint depression that can be measured is limited to 36.5 K at 0 °C and 12.6 K at –80 °C. This corresponds to measurement limits of 3%–6% RH

(Vömel et al., 2003). In some cases, loss of the frost point control within layers with RH below this detection limit has led to inaccurate measurements even above these dry layers where the RH is within the detection range of the instrument (Vömel et al., 2003). In the stratosphere, the mirror temperature often indicates higher-than-expected frost point temperatures even within the cooling limit (Sugidachi, 2014).

Continuous accurate measurements of water vapor are essential for climate monitoring. The Global Climate Observing

System (GCOS) Reference Upper-Air Network (GRUAN) requires monthly water vapor profiles up to ~30 km (WMO, 2013). Although the CFH and NOAA FPH are currently considered the most reliable instruments for climate monitoring purposes, their use is increasingly reduced as a result of the restrictions on the production and application of $CHF_3$ in various countries. The demand for an environmentally friendly, CHF3-free instrument spurred the development of a Peltier-based digitally controlled chilled-mirror hygrometer named "SKYDEW" to measure atmospheric water vapor without using a

cryogen (Sugidachi, 2011, 2014). In this paper, we present the technical details, processing algorithms, and uncertainty estimates of SKYDEW measurements derived from soundings at various latitudes.

The remainder of this paper is organized as follows. Section 2 gives a description of the SKYDEW instrument, and Section 3 describes the data processing. Section 4 assesses the uncertainty, and the results of intercomparisons with other instruments used for verification are described in Sect. 5. Finally, a summary is provided in Sect. 6.




## 2 Instrument

### 2.1 Instrument description

Figure 1 shows a schematic drawing together with a photograph of the SKYDEW hygrometer. The mirror is a thin silicon wafer of size 2 mm × 2.5 mm × 0.28 mm. A platinum resistance thermometer (PT100) is located between the mirror and the

cold side of a Peltier device. A two-stage Peltier device is used that allows greater cooling capacity than a single stage device such as used for Snow White. Due to this larger cooling power, at temperature difference of more than 90 °C can be achieved at room temperature. SKYDEW measures the intensity of the scattered light from the mirror to monitor the condensate (dew/frost) amount. Therefore, the surrounding of the mirror is open with no obstructions, as the photodetector and the light source are at the same location. The light source is modulated to reduce interference from sunlight, similar to

the CFH and FPH. The instrument weighs less than 500 g, including dry-cell batteries (one 006P type battery and six AA type batteries). For a previous version of the instrument ethanol was used for additional cooling of the heatsink, but since June 2020 a larger heatsink was implemented to cool the hot side of the Peltier device, eliminating the need for ethanol cooling, which makes it easier to prepare and handle the instrument.

SKYDEW monitors the mirror temperature, the intensity of the scattered light, the current through the Peltier device,

the heatsink temperature, and the battery voltage at a rate of 1 Hz. Except for the mirror temperature these parameters are used as housekeeping data to monitor whether the system is working properly. The instrument is not equipped with a transmitter so that it relies on a radiosonde to transmit the data together with the parameters that are usually measured by an operational radiosonde (e.g. temperature, RH, pressure, altitude and position information from the global positioning system (GPS)). SKYDEW is compatible with the Meisei RS-11G, using a Meisei-specific data format, or with any other radiosonde

that supports Xdata (such as Vaisala RS41 Intermet 54, Graw DFM-17), which is a common protocol for transferring data to radiosondes (Wendell and Jordan, 2016).

### 2.2 Calculation of relative humidity and mixing ratio from mirror temperature

The mirror temperature is expected to correspond to the dew/frost point temperature when equilibrium is established

between vapor and liquid water/ice. The dew/frost point temperature can be converted to vapor pressure using solutions of the Clausius–Clapeyron equation, which provides an expression for the boundary curve of phase equilibrium in the pressure–temperature (*p*–*T*) plane, as follows:

$$\frac{d\,\ln p}{dT} = \frac{L(T)}{RT^2},$$

(1)

where $p$ is vapor pressure, $L(T)$ is latent heat as a function of temperature $T$, and $R$ is the specific gas constant of water vapour (461.5 J kg$^{-1}$ K$^{-1}$). Several empirical equations based on experimental data have been developed for the relationship between temperature and water vapor pressure (Murphy and Koop, 2005; Vömel 2016). The saturation water vapor pressure $e_w$ of pure water vapor with respect to liquid water is the partial pressure of the vapor when in a state of equilibrium with a





plane surface of pure liquid water at the same temperature and pressure. Similarly, $e_i$ stands for the water vapor pressure

with respect to ice. $e_w$ and $e_i$ are functions of temperature only:

$$e_w = e_w(T),$$
$$e_i = e_i(T).$$

(2)

Although $e_w$ and $e_i$ need to be corrected for the mixture of dry air and water vapor using an enhancement factor $f(T, p)$ to

obtain the saturation vapor pressure $e'_w$ and $e'_i$ of moist air, in the meteorological range of pressure and temperature, $e'_w$ and $e'_i$ are almost equal to $e_w$ and $e_i$, respectively, with an error of ≤0.5% (Buck, 1981; Murphy and Koop, 2005). For measurements of tropospheric and stratospheric water vapor, this enhancement factor is ignored (Vömel et al., 2007). In chilled-mirror hygrometers, the phase of the condensate on the mirror may introduce an ambiguity; it is unclear whether the mirror temperature is the frost point or dew point temperature if the phase is not known (Fujiwara et al., 2007). Below 0 °C,

liquid water has a vapor pressure lower than that of ice. Therefore, the frost point is higher than the dew point under the same vapor pressure conditions. At –20 °C, where supercooled water may exist on the mirror, the temperature difference between the dew and frost points reaches ~2 K. Experiments with Snow White show that the condensate on the mirror can be super-cooled (liquid) water for temperatures as low as –25 °C to –30 °C. The CFH control algorithm eliminates this ambiguity by forced freezing, i.e., boiling off the condensate followed by forcing the mirror to cool down to below –38 °C,

which quickly freezes any liquid water on the mirror. This procedure is executed when the measured frostpoint temperature reaches –12.5 °C and -55C. This forced freezing allows the clear identification of the liquid-to-ice transition (Vömel et al., 2007).

We convert the vapor pressure data to other forms of water vapor concentration (e.g., RH and volume mixing ratio), by combining other simultaneous radiosonde measurements such as temperature and air pressure. We use the equation of

Hyland and Wexler (1983) for the temperature–water vapor pressure relationship, which is one of the equations recommended by Nash et al. (2011). We calculate the RH by using the following equations (Fujiwara et al., 2003). If the condensate on the mirror is liquid water (dew):

$$\text{RH} = \frac{e_w(T_\text{m})}{e_w(T_\text{air})} \times 100\%,$$

(3)

where $T_\text{m}$ is the mirror temperature of SKYDEW and $T_\text{air}$ is the air temperature. If the condensate on the mirror is ice (frost):

$$\text{RH} = \frac{e_i(T_\text{m})}{e_w(T_\text{air})} \times 100\%.$$

(4)

WMO (2008) recommends the use of RH with respect to liquid water even at temperatures below 0 °C. All radiosondes employed at WMO stations report the RH with respect to liquid water. However, in the upper troposphere/lower stratosphere

(UT/LS) science community, RH with respect to ice, $\text{RH}_i$, is also used as follows:



$$\mathrm{RH_i} = \frac{e_i(T_\mathrm{m})}{e_i(T_\mathrm{air})} \times 100\%.$$

(5)

The volume mixing ratio $\chi$ [ppmv] is calculated by dividing the water vapor partial pressure by the pressure of dry air, $P$, as follows:


$$\chi = \frac{e}{P} \times 10^6.$$

(6)

For the specific humidity and precipitable water, please refer to the National Weather Service website (https://www.weather.gov/lmk/humidity).

### 2.3 Cooling capacity of the Peltier device


The SKYDEW mirror is cooled by a two-stage Peltier device. Figure 2 shows the cooling capacity measured in an environmental chamber, which indicates the dependence on air temperature, air pressure, and flow rate near the mirror. The cooling capacity is calculated from the heat budget on the mirror by considering the following factors: the heat transfer by the Peltier effect, the resistive heating, and heat exchange with ambient air (Sugidachi, 2014). The temperature difference $\Delta T$ created by a Peltier device is given by:


$$\Delta T = \frac{R_e I^2/2 + \alpha I T_\mathrm{air}}{-\alpha I + \beta + HS},$$

(7)

where $R_e$ [ohms] is the resistance of the Peltier device, I [A] is the current through the Peltier device, $\alpha$ [V K$^{-1}$] is the coefficient derived from the material and the size of the Peltier device, $T_\mathrm{air}$ is air temperature, $\beta$ [W K$^{-1}$] is the coefficient associated with the thermal conductivity of the Peltier device, $H$ [W m$^{-2}$ K$^{-1}$] is the heat transfer coefficient between the


mirror surface and air temperature, and $S$ [m$^2$] is the mirror surface area. Under the assumption of laminar flow, $H$ may be written as follows:

$$H = 0.0143 U \rho \left(\frac{\kappa}{\mu}\right)^{\frac{2}{3}} c_a^{\frac{1}{3}},$$

(8)


where $U$ is air flow [m s$^{-1}$], $\rho$ [kg m$^{-3}$] is air density, $\kappa$ [W K$^{-1}$] is thermal conductivity, $\mu$ [kg m s$^{-1}$] is the viscosity of air, and c$_a$ [J g$^{-1}$ K$^{-1}$] is the heat capacity of air (Sakata, 2005). According to Eq. (7) the temperature difference $\Delta T$ increases with increasing Peltier current, it is counter acted by resistive heating if the current gets too large, meaning that the cooling efficiency will decrease when the current exceeds ~2 A.

Equations (7) and (8) describe quantitatively the performance of the Peltier device depending on the environmental


conditions. For higher temperature and lower pressure, $\Delta T$ increases. Based on the results of the chamber experiments shown



in Fig. 2 together with equations (7) and (8) we estimate that the maximal achievable temperature difference $\varDelta T$ is >55 K at the surface, which corresponds to ~1% RH at 25 °C, and close to 30 K under the least favourable conditions, at a temperature of -70 °C. The Peltier device creates a temperature difference between the cold side (mirror) and the hot side (heatsink). Therefore, it is important that the heat sink is efficient and that it remains at, or close to, ambient temperature. In case of a large dew point depression (dry air) high cooling power needed to cool the mirror, and the efficiency of the heatsink to get rid of this excess energy determines the performance of the SKYDEW instrument.

### 2.4 Thermometer for mirror temperature measurement

The mirror temperature is measured with a platinum resistance temperature detector (PT100) that which has a unique repeatable and predictable resistance–temperature (R–T) relationship. The unique properties of platinum make it the material of choice for temperature standards (Immler et al., 2010). We use the four-wire configuration to measure the resistance of the PT100. The PT100 was calibrated at four temperatures (40 °C, 0 °C, –45 °C, and –95 °C) to characterize the individual $R$–$T$ relationship of the PT100. We have also calibrated the constant current circuit and AD (Analog–Digital) converter individually to measure the resistance of the PT100 accurately. Figure 3 shows the $R$–$T$ relationship and the errors from the calibration curve at each calibration point. The values are the averages for 20 PT100 sensors. The maximum uncertainty is ~0.05 K ± 0.02 K. In addition to the temperature calibration, the calibration for resistance measurement is conducted at 55, 70, 85, 100, 115 ohms (corresponding to –110 °C to 40 °C) to measure the resistance of PT100. The possible error of this process is <0.005 K.

### 2.5 Feedback Controller

A PID controller is used as the feedback controller to maintain the equilibrium of the condensate amount/size. The PID controller calculates an output value by using an "error" value as the difference between a measured process value and a desired set point. The controller attempts to minimize the error by adjusting the process control inputs (Åström and Murray 2008). In SKYDEW, the measured process value is the scattered light intensity from the condensate, the desired set point is the desired scattered light intensity (see Sect. 2.6 for details), and the control outputs correspond to the current of the Peltier device. The error refers to the difference between the desired scattered light intensity and the current value. In general, the control output of the PID controller is calculated as follows:

$$u(t) = K_p \left\{ e(t) + \frac{1}{T_{\text{int}}} \int_0^t e(\tau)d\tau + T_{\text{der}} \frac{de(t)}{dt} \right\},$$

(9)

where $u(t)$ is the control output, $e(t)$ is the error, $K_p$ is a proportional gain constant (P), $T_{\text{int}}$ is the integration time (I), and $T_{\text{der}}$ is the differentiation time (D). A high $K_p$ results in a large change in the output. Increasing $K_p$ will result in the feedback control becoming highly responsive and unstable. Decreasing $K_p$ will result in the feedback controller becoming





stable with low responsivity. Only a proportional control cannot eliminate the residual error, as it requires an error to

generate a proportional output. In other words, an offset error will remain. The integral term eliminates this residual error by

integrating the error over time to produce an "I" component for the controller output. The derivative term predicts the system

behaviour and thus reduces the settling time.

It is necessary to find the best set of PID parameters (i.e., $K_p$, $T_\mathrm{int}$, and $T_\mathrm{der}$) to achieve optimal control, which may

depend strongly on the dew/frost point temperature. The PID parameters for a chilled-mirror hygrometer need to be adjusted

for various atmospheric conditions by gain scheduling (Vömel et al., 2007a; Hall et al., 2016). Physically, the feedback

control on a chilled-mirror hygrometer is related to the speed of deposition and evaporation of condensate, which depend on

the degree of supersaturation around the condensate on the mirror. It is a reasonable assumption that the PID parameters

should vary with the water vapor pressure; i.e., the mirror temperature. As a result of the chamber test, we found that the

value of $K_p$ should be larger at low mirror temperatures. Therefore, the value of $K_p$ is a function of the mirror temperature,

and is updated every second during the sounding whereas $T_\mathrm{int}$ and $T_\mathrm{der}$ are kept constant.

## 2.6 Condensate on the mirror

### 2.6.1 Phase transition

It is important to know the phase of the condensate on the mirror. Sugidachi (2014) investigated the condensate on the mirror

during the phase transition from water to ice using an optical microscope. Figure 4 shows the behaviour of the phase

transition at an air temperature of ~0 °C.

The condensate consists of 5–10 µm water droplets (supercooled water) for about 500 s ((1) in Fig. 4) after the start of

the cooling. At t=800 s, ice starts to form at the upper right edge of the mirror (2). The ice crystals grow gradually, while the

number of supercooled water droplets decreases (i.e., they evaporate) (3). After a few hundred seconds, the supercooled

water on the mirror has completely disappeared (4). It is thought that the mirror temperature for phases (1)–(3) is colder than

the frost point; consequently, the ice on the mirror grows. Compared with the dew point and frost point temperature derived

from coincident radiosonde RH and temperature data, the mirror temperature seems to indicate the dew point for phases (1)–

(3), and the frost point for phase (4). The experimental results show that the mirror temperature is the dew point temperature

when the condensate on the mirror is mixed phase. Under conditions (1)–(3), the scattered light shows large fluctuations (~

±0.02 V s$^{-1}$) and the period of oscillation is shorter than that for condition (4). This implies the output of PID controller is

mainly dominated by the change in the water droplets on the mirror. During this time, a phase change from water to ice

occurs, which requires the mirror's temperature to be higher than the dew point. However, the phase transition is quite slow,

so the difference of mirror temperature from the dew point is small and can be considered negligible. Therefore, we can

determine whether the mirror temperature represents the dew point or the frost point temperature from the behaviour of the

scattered light; i.e., the mirror temperature is the dew point temperature when the scattered light shows large and rapid

fluctuations, and the mirror temperature is the frost point temperature when the fluctuations are small and slow, as shown by

the blue curve on the top panel in Fig. 4. These different behaviours are because ice and water have different evaporation



rates, but are controlled by the same PID controller settings. For the CFH and NOAA FPH, the forced freezing algorithm is applied at a mirror temperature of –12.5 °C to eliminate the phase ambiguity.


### 2.6.2 Ice size at low temperatures

The condensates on the mirror surface were investigated in a thermostatic chamber (Sugidachi, 2014). Figure 5 shows the condensates on the mirror. The large ice crystals that formed from supercooled water by the phase transition are maintained at low temperature (about –30 °C) for 20–30 min. We evaporated these large ice crystals by heating the mirror once and re-

formed the condensates under sufficiently cold temperatures of –20 °C, –40 °C, and –60 °C. A larger number of smaller ice crystals formed at lower temperatures. For SKYDEW, the intensity of the detection signals corresponds to the backscattering by the condensates on the mirror, which is thought to depend on the particle size and the wavelength of emitted light, assuming the backscattering is Mie scattering (Craig and Donald, 1983; Fujiwara et al., 2016). The relationship can be approximated by a monotonically increasing function. Light scattered by a small particle gives weak signals. Because the

desired level of the scattered light intensity was set to a constant 1.3 V throughout this experiment, more ice particles should be formed on the mirror when the particle size is small; i.e., under cold conditions. For cloud microphysics, the number of ice-forming nuclei (IN) increases nearly exponentially with decreasing temperature (Pruppacher and Klett, 1997). As in the cloud formation process, the temperature dependence of condensate number on the mirror may reflect the temperature dependence of the IN on the mirror.

The size of the ice crystals on the mirror affects the response time. Let us estimate the evaporation rate of the condensate on the mirror, using the equation for the evaporation rate of a droplet falling from the cloud base (Pruppacher and Klett, 1997). In the case of a falling droplet, the evaporation rate can be calculated as follows:

$$\frac{dr}{dt} = \frac{D_v}{r\rho_w \mathrm{R}}\left(\frac{e(T_\infty)}{T_\infty} - \frac{e(T_r)}{T_r}\right)\bar{f}_v,$$

$$\bar{f}_v = 1.00 + 0.108\left(\mathrm{Sc}^{\frac{1}{3}}\mathrm{Re}^{\frac{1}{2}}\right)^2,$$

315 (10)

where $r$ is the radius of the droplet, $D_v$ is the diffusion coefficient, $\rho_w$ is the density of water, R is the gas constant for water vapor, $e(T_\infty)$ is the vapor pressure for ambient air temperature $T_\infty$, $e(T_r)$ is the vapor pressure for the drop surface temperature $T_r$, and $\bar{f}_v$ is the ventilation coefficient. The Schmidt number $\mathrm{Sc} = \frac{\mu}{\rho D_v}$, where $\mu$ is the dynamic viscosity and $D_v$ is the mass diffusivity. The Reynolds number $\mathrm{Re} = \frac{\rho v L}{\mu}$, where $L$ is a characteristic linear dimension and $v$ is the

kinematic viscosity. Assuming that the condensate on the mirror consists of spherical ice crystals, the evaporation rate can be calculated as follows:

$$\frac{dr}{dt} = \frac{\mathrm{D}_v}{r\rho_w \mathrm{R}_v}\left(\frac{e(T_{\mathrm{fp}})}{T_{fp}} - \frac{e(T_m)}{T_m}\right)\bar{f}_v,$$



(11)

where $T_{fp}$ is the expected frost point temperature in the stratosphere and $T_m$ is the mirror temperature. The evaporation rate

depends on the crystal radius and the vertical gradient of the water vapor density near the mirror surface. Using Eq. (11), it

follows that the condensate remains on the mirror for nearly ~20000 s under conditions of the mirror temperature of -70 °C

and the frost point temperature of -80°C if its size is larger than ~20 μm, whereas the small particles (~2 μm) evaporate

rapidly (~200 seconds).

To ensure a fast response during the ascent, the size of the ice condensate particles must be kept small. This is achieved

by applying a five second long heat pulse at sufficiently low mirror temperature to evaporate the condensate and to

subsequently force the formation of a condensate of small ice droplets

The heating/forced freezing procedure is conducted two times during a sounding. The first heating is applied at a mirror

temperature of about –36 °C and the second heating is at a mirror temperature of about –58 °C. These temperatures are a

compromise between rebuilding the condensate at as low as possible temperature to maintain a fast response time, and the

increasingly longer time needed to form ice at lower temperatures, and are based on the Peltier's cooling performance.

The amount/size of water or ice formed on the mirror surface is determined by the desired set point of scattered light in

the PID controller. To obtain a fast response, it is desirable that the amount of water and ice on the mirror surface is small.

The evaporation rate at low temperatures is slow and the variation of scattered light is small, so stable control is possible

even at a lower set value of the desired scattered light intensity.

The scattered light intensity from a clean mirror with no cooling and no water/ice formation sets the base level. The

base level of the scattered light signal varies due to individual differences and is determined during factory calibration. In

SKYDEW, the desired scattered light intensity is set to +0.3 V above the base level of scattered light intensity until the first

heating (i.e., when the mirror temperature exceeds about –36 °C). After the first heating, the desired scattered light intensity

is set to +0.25 V higher than the base level, and after the second heating (i.e., when the mirror temperature is less than about

–58 °C), it is set to +0.2 V higher than the base level.

## 3 Data processing

The PID controller of balloon-borne chilled-mirror hygrometers is tuned to respond quickly to changes in water vapor in the

atmosphere. This tuning often produces small oscillations around the true dew/frost point in measured mirror temperatures.

For the final data product, these oscillations should be smoothed by an appropriate filter. Most conventional chilled-mirror

hygrometers apply simple filtering such as a gaussian filter or moving average. Poltera (2022) and Poltera et al. (2021)

proposed the so-called golden point method, which is a smoothing method based on the measurement principle of the

chilled-mirror hygrometer. Here, the golden point means the equilibrium points when the condensate on the mirror neither

grows nor shrinks. In this section, we describe the data processing for the SKYDEW measurements including the golden

point method.





### 3.1 Quality control

Before smoothing, the mirror temperatures during the intentional heating stages and under the Peltier cooling limit should be eliminated. As described in Sect. 2.6.2, intentional heating is carried out at –36 °C and at –58 °C for 5 s each time to reform

ice crystals on the mirror. The mirror temperature during these heating periods is not a measure of the dew/frost point. Figure 6 shows an example of the heating control. When the mirror temperature reaches up to about –58 °C, the mirror is heated up to –20 °C and the scattered light signal indicates the base level (i.e., no ice on the mirror). After 5 s, the mirror temperature is cooled down again and the scattered light signal returns to the desired value after a small overshoot. The mirror temperatures during the heating and overshooting are removed during data processing. SKYDEW cannot take measurements under

extremely dry conditions (dewpoint depression > 60 K at the surface) because of the cooling limit derived from the Peltier performance. The scattered light signal is used to judge whether a constant dew/frost layer is retained. If the scattered light signal continues to lie below the desired value, the mirror temperatures are higher than the dew/frost point and no condensate is present. This situation is often observed above an altitude of 20–25 km for daytime soundings and above 25 km for nighttime soundings. The altitude where the scattered light signal starts to fall below the desired values is the measurement

limit for SKYDEW, and therefore the mirror temperatures at higher altitudes are labelled as missing.

### 3.2. Smoothing by the golden point method

Golden points for chilled-mirror hygrometers are the equilibrium points of dew/ice on the mirror, which correspond to the extrema of the scattered light signal. The scattered light signal on the PID controller oscillates around the desired value, and

maxima and minima points can be extracted from the periodic oscillation of the scattered light signal as shown in Fig. 7.

       The output data rate for SKYDEW is 1 Hz. This resolution is slightly too low for detecting equilibrium points for ice on the mirror. Linear interpolation is therefore applied to produce 5 Hz data and fill missing data points. Next, we remove random electrical noise from the scattered light signal and mirror temperature to avoid interference when extracting the extrema. The window of the smoothing filter should be appropriately chosen to remove only electrical noise, not the

oscillations of the PID controller. Here we use a gaussian filter, with sigma (σ) varying with mirror temperature, between 1.0 s at a mirror temperature of 50 °C and 3 s at –100 °C. Then, the extremum of the scattered light (both maximum and minimum) is detected from the smoothed scattered light and the mirror temperature corresponding to the extremum is extracted. Finally, the extracted points are smoothed with a gaussian filter with σ = 1.5.

       The vertical interval between the golden points is generally longer at higher altitude. Figure 7 shows an example of a

profile of dew/frost point determined by the golden point method in the troposphere (bottom panel) and stratosphere (upper panel). In the lower troposphere (stratosphere), the oscillation amplitude is large (small) and the period is short (long) because the growth and evaporation rates of the ice are high (low).




## 4 Uncertainty estimation

In this section, we estimate the measurement uncertainties in the processed SKYDEW data product, following the guidelines in JCGM/WG1 (2008). There are several sources of uncertainty in processed data measured with a chilled-mirror hygrometer, which may be divided into technical and physics factors. Technical factors are the mirror temperature measurement (Sect. 4.1), the stability and response of the PID controller (Sect. 4.2, 4.3), and the contamination error due to cloud/rain droplets (Sect. 4.4). Physics factors include the ambiguity of the condensate phase (Sect. 4.5), aerosol effects and

the curvature effect (Sect. 4.6). In addition, the uncertainty of the vapor equation should be considered in the conversion to vapor pressure (Sect. 4.7). These uncertainties are discussed in more detail in this section.

### 4.1. Mirror temperature measurement

Chilled-mirror hygrometers should be fundamentally calibrated by a humidity standard because they measure humidity.

However, there is no humidity standard under the low-pressure and low-temperature conditions found in the upper air. Therefore, we took the traceability for the temperature measurement instead of humidity, as is done for the CFH (Vömel et al., 2016) and NOAA FPH (Hall et al., 2016) under the assumption that the mirror temperature is equal to the dew/frost point temperature.

SKYDEW uses a platinum resistance thermometer (PT100) as described in Sect. 2.4. The maximum error value is

~0.05 K ± 0.02 K (Fig. 3). Here, we assume that the true value exists within a continuous uniform distribution of +/–0.07 °C. By dividing this value by $\sqrt{3}$ to reduce this distribution to its standard deviation equivalent, the standard uncertainty from the temperature calibration is $0.07/\sqrt{3}$. In addition to the temperature calibration, the calibration for resistance measurement is conducted at 55, 70, 85, 100, 115 ohms (corresponding to –110 °C to 40 °C) to measure the resistance of PT100. The possible error on this process is <0.005 K. The thermal gradient of the mirror is estimated within 0.015 K from the thermal

conductivity. These uncertainties derived from temperature measurements are mainly systematic uncertainties and are correlated within a single profile and for long time series.

### 4.2. Error of golden point selection

Errors in golden point selection lead to errors in dew/frost point estimation, as shown in Fig. 8. The error of the estimated

dew/frost point depends on the oscillation period, amplitude, and the noise level of scattered light. Here, the uncertainty is estimated under the assumption that the timing error for golden point selection is ≤1 s. Smaller amplitudes of oscillations result in better estimation of dew/frost point. Shorter oscillation periods result in higher vertical (time) resolution. The standard uncertainty from the golden point error varies greatly depending on the amplitude of the oscillation, and has values in the range 0.1–0.5 K. The uncertainty from the golden point error originally derives from electrical noise on the control

circuit board. Therefore, this uncertainty is random and uncorrelated.



### 4.3. Filtering the deviation error

A gaussian filter is used for smoothing the selected golden points. The associated uncertainty is expressed as the weighted standard error of the mean (Gatz and Smith, 1995; Vömel et al., 2016) as follows:

$$\delta_{\overline{T}_i}^2 = \frac{n}{n-1}\left[\sum_j \left(w_{i,j}T_j - \overline{w_i}\,\overline{T_i}\right)^2 - 2\overline{T_i}\sum_j \left(w_{i,j} - \overline{w_i}\right)\left(w_{i,j}T_j - \overline{w_i}\,\overline{T_i}\right) + \overline{T_i}^2 \sum \left(w_{i,j} - \overline{w_i}\right)^2\right],$$

(12)

where $n$ is the number of 5 Hz data points in each filter window, $T_j$ is the mirror temperature at time-step $j$, and $w_{i,j}$ is the weight at time-step $j$ to calculate the value of the filtered time series at time-step $i$. This uncertainty is related to the error of golden point selection and therefore is uncorrelated for time series. The typical value is >0.5 K when oscillations are large.

### 4.4 Contamination by outgassing from the balloon surface and payload

Contamination due to outgassing from the balloon surface and cloud/rain droplets affects the dew/frost point measurements (Hall et al., 2016; Vömel et al., 2016; Jorge et al., 2021). Kräuchi et al. (2016) showed an example of stratospheric contamination on an ascent profile starting from 25 km, with uncontaminated measurements above 27 km on the controlled descent measurement. Avoiding the contamination errors due to trapped cloud water droplets is the greatest challenge for balloon borne stratospheric water vapor measurements during ascent. We have minimized the contamination from the payload by designing the mirror part of the sensor to be at the top of the payload, and tried to reduce the contamination from the balloon surface by separating the instrument from the balloon using an unwinder longer than 30 m. However, unrealistic mixing ratios are sometimes observed above ~25 km which is attributed to contamination by the balloon wake. Since the instrument is working properly, but measuring water vapor from an artificial source, it is not possible to estimate the contamination error from the scattered light signal or from analysing SKYDEW housekeeping data alone. The contamination can be estimated by comparison with other sensors such as collocated satellite data or even the climatology. Descent data, even if uncontrolled, are also useful for assessing the contamination during ascent. Note that the descent SKYDEW data are not always available because SKYDEW is sometimes out of control due to the cooling limit in the stratosphere.

### 4.5 Ambiguity of the condensate phase

Below mirror temperatures of 0 °C, there is ambiguity in the condensate phase (Vömel et al., 2016; Fujiwara et al., 2003; Murphy and Koop 2005). This ambiguity can be eliminated by monitoring the scattered light fluctuations, as discussed in Sect. 2.6.1. For SKYDEW, the phase transition occurs between 0 °C and –36 °C mirror temperature. Comparison with the simultaneously flown radiosonde RH sensor is also useful to determine whether the mirror temperature is the dew point or frost point temperature (Fujiwara et al., 2003).





Murphy and Koop (2005) suggested that at very cold temperatures (T < 200 K) water ice may exist not only in its normal hexagonal crystal structure but also with a cubic crystal structure. The vapor pressure of the cubic ice is 3%–11%

higher than that of hexagonal ice, which would cause a higher mirror temperature for frost point hygrometer measurements. The CFH and NOAA FPH controllers operate so as to evaporate and immediately reform the frost layer at –53 °C (Vömel et al., 2007; Hall et al., 2016). At this temperature only hexagonal ice will form from water vapor. In this way, the frost layer is maintained into the colder region and will remain in the hexagonal phase. The heating control at –58 °C for SKYDEW prevents the formation of cubic ice while also giving a faster response by forming small ice particles on the mirror.


### 4.6 Aerosol effect and curvature effect

Dissolved pollutants affect the vapor pressure (Thornberry et al., 2010). If the condensates are formed by solutions, the vapor pressure is smaller than that of pure water (the so-called Raoult's law). We have set the mirror angle in SKYDEW obliquely downward (30° to the vertical direction) to avoid cloud droplets and aerosol particles attaching to the mirror. Also, gases

other than water vapor may interfere with the frost point temperature measurement. For the UT/LS region, $HNO_3$ is adsorbed readily onto the ice surfaces to form nitric acid hydrates such as nitric acid tri-hydrate (NAT), which would increase the apparent frost point temperature (Szakáll et al., 2001). However, experimental results by Thornberry et al. (2010) show that no detectable interference in the measured frost point temperature was found for $HNO_3$ mixing ratios of up to 4 parts per billion (ppb). The actual amount varies from <0.1 ppb to several ppb in the UT/LS. Therefore, the aerosol effect should be

negligible.

We may need to consider the curvature effect (or the surface tension effect) when calculating the vapor pressure from the dew/frost point temperature, when the condensate size is small (Pruppacher, 1997; Murphy and Koop, 2005). The size of the condensate on the mirror in the liquid phase is 5–10 μm, and that of the ice particles on the mirror is 2–5 μm at a temperature of –60 ˚C. The saturation vapor pressure over ice particles is written as

$$\frac{e_i(r)}{e_i(\infty)} = \exp\left(\frac{2\sigma}{r\rho_w \mathrm{R}T}\right),$$

(13)

where $e_i(r)$ is vapor pressure over ice with radius $r$, and $\sigma$ is surface tension (Pruppacher and Klett, 1997). Using $\sigma = 106 \pm 3 \times 10^{-7}$ J cm$^{-2}$ (Pruppacher and Klett, 1997) and $r = 1$ μm, we obtain $e_i(r)/e_i(\infty) \approx 0.002$. Therefore, the particle size contributes less than ~0.02 K ($\{e_i(\mathrm{T}+0.02) - e_i(\mathrm{T})\}/e_i(\mathrm{T}) \approx 0.002$) to the frost point temperature measurement.


### 4.7 Other sources of uncertainty

If we calculate RH and mixing ratio by volume, the uncertainty of the air temperature and pressure measurements also need to be considered. SKYDEW is typically connected to the Meisei RS-11G or Vaisala RS41 radiosonde, whose measurement uncertainty is fully described by Kizu et al. (2018) and Sommer et al. (2023), respectively. The combination with these

GRUAN data products (GDPs) contributes to the uncertainty of the SKYDEW RH and mixing ratio.





## 4.8. Total uncertainty

The total uncertainty is estimated by combining calibration errors (Sect. 4.1), the golden point determination error (Sect. 4.2), and the filtering standard error (Sect. 4.3). An example of a SKYDEW uncertainty profile is shown in Fig. 9(d). The

uncertainty from the filtering and golden point determination errors dominates the uncertainty. These terms may be considered random and uncorrelated uncertainties, which are unlikely to impact long-term trends. The uncertainty regarding the mirror temperature measurement is a small but systematic error that is correlated within a single profile and in the long time series. Therefore, this uncertainty should be considered carefully when detecting long-term trends with SKYDEW data. The uncertainty estimate does not include the contamination error because we cannot estimate this effect with ascent data

alone from SKYDEW; this will be discussed in the next section. Finally, Table 1 gives a list of uncertainty sources and values.

RH is calculated with dew/frost point temperature and air temperature from a radiosonde. Therefore, the uncertainty of RH is calculated using the law of propagation of uncertainty. The combined standard uncertainty of RH, $u_{RH}$, is written as

$$u_{RH} = \sqrt{\left(\frac{\partial RH}{\partial T}\right)^2 u_T^2 + \left(\frac{\partial RH}{\partial Td}\right)^2 u_{DP}^2},$$

500                                                                                                                               (14)

where $u_T$ is the uncertainty of air temperature, and $u_{DP}$ is the uncertainty of the dew/frost point described in Table 1. Figure 9(e) shows the RH profile and Fig. 9(f) its uncertainty for this sounding. The RS-11G, which is a radiosonde with a certified GDP, was used with SKYDEW to obtain the air temperature and pressure. Therefore, the uncertainty in the GDP is available as $u_T$. The RH uncertainty is generally <1.5% RH for the whole profile, expect for the region with rapid RH changes, where

the dew/frost point uncertainty is large.

The water vapor mixing ratio (WVMR) is calculated by combining with pressure. The combined standard uncertainty of WVMR is written as

$$u_{RH} = \sqrt{\left(\frac{\partial WVMR}{\partial P}\right)^2 u_P^2 + \left(\frac{\partial WVMR}{\partial Td}\right)^2 u_{DP}^2},$$

(15)

where $u_p$ is the uncertainty of air pressure. Figure 9(g) shows the WVMR profile and Fig. 9(h) its uncertainty. The uncertainty of WVMR is in the range 0.25–1.0 ppmv, depending strongly on the value of $u_{DP}$. The uncertainty of pressure is very small and has an almost negligible effect on the combined uncertainty.

## 4.9 Golden point method vs gaussian filtering

Figure 10 shows the dew/frost point profiles at Tateno on 26 November 2021 processed with both a normal gaussian filter (green) and the golden point method (red), as well as the difference between the two profiles (black). The sigma (σ) of a





gaussian filter varies with mirror temperature, between 0.5 s at a mirror temperature of 50 °C and 20 s at –100 °C. The uncertainty is calculated with Eq. (12). The uncertainty derived from the temperature calibration is the same as in Section 4.1. The uncertainty when the data are processed with the gaussian filter is roughly 0.3–0.4 K, as shown in Fig. 10b. The

uncertainty of the gaussian filter is larger than that of the golden point method in the range 5–25 km. Figure 10c focuses on the profiles around 9 km, where there are sharp gradients in frost point. Gradients in the profile from the golden point method are sharper than those from the gaussian filter, and the golden point method can detect the peak at ~8.9 km. In addition, the profile from the gaussian filter retains small oscillations around 9.5 km, where strong oscillations occur as a result of aggressive PID control. On the other hand, there is no oscillation for the profile using the golden point method.

The frost point temperature from the gaussian filter tends to be slightly lower than that from the golden point method in the stratosphere. This may be due to an imbalance of the cooling and heating power of the Peltier method and different magnitudes of the growth/evaporation rate of ice on the mirror. It is assumed that the oscillations are usually symmetric around the true frost point temperature. However, in the case of strong oscillation of mirror temperature, there is a possibility that the oscillation is not centred at the true frost point because of the difference between the evaporation and condensation

rate (Vömel et al., 2016). However, the difference is ~0.1 K at most, even in the stratosphere (Fig. 10a).

## 5 Verification

This section provides sounding results to evaluate the measurement performance of SKYDEW. Since 2011, we have conducted over 40 test soundings in various parts of the world, including tropical regions in Indonesia, mid-to-high latitude

regions in Japan and Germany, and over the Pacific Ocean. Among these soundings, dual soundings are useful for the verification of reproducibility including the sensor calibration and individual differences of cooling control. Some intercomparisons with other instruments were also conducted to evaluate the measurement performance.

### 5.1 Dual sounding for reproducibility evaluation

Dual sounding with two SKYDEWs was conducted on 4 April 2019 at Lindenberg, Germany, to evaluate reproducibility (Fig. 11). This version of SKYDEW had a smaller heatsink so that the cooling limit is lower than in the latest version, but apart from that there are no significant differences in the hardware and software between both versions. We analysed the dew/frost point from the surface to 16.5 km, where both SKYDEWs work correctly. The combined standard uncertainty is calculated with two SKYDEW profiles in each 100 m vertical bin according to the law of uncertainty propagation. The

difference between the two SKYDEWs is almost within these uncertainties ($k = 2$) up to 16.5 km. As mentioned in Sect. 4, the systematic error for SKYDEW is <0.1 K, and therefore this sounding result shows no systematic bias. Most of the uncertainties are random, related to the golden point estimation. The uncertainty tends to be large during strong oscillations





of mirror temperature. Smaller oscillation is required for better measurement, although the oscillation due to the aggressive setting of the PID controller is needed to detect the golden point.


## 5.2 Comparison with CFH

The comparisons with CFH were conducted several times in Japan, Germany, and Indonesia. In this section, we describe the results of two soundings, at Tateno in Japan and Lindenberg in Germany.

Figure 12 shows the result at Tateno on 20 April 2018. This sounding was conducted using two separate balloons, one
for SKYDEW and the other for the CFH, but released at the same time. SKYDEW was able to measure the dew/frost point up to 21 km, which was the cooling limit altitude for SKYDEW for this case. This sounding was conducted during daytime, so that the cooling limit was reached at a lower altitude than for a nighttime observation. The difference between CFH and SKYDEW, averaged for each 100 m bin, is roughly less than the uncertainty ($k = 2$) in the stratosphere. Here, we assume that the CFH uncertainty is 0.2 K through the whole profile (Vömel et al., 2016). The deviation is large at the troposphere
because the CFH and SKYDEW are released separately with different balloons, so they do not sample the same air. In the stratosphere, there appears to be systematic differences within uncertainties ($k = 2$).

Figure 13 shows the profiles from SKYDEW and CFH using a single balloon on 1 April 2019 at Lindenberg and the difference between the two measurements. The values are averaged over each 100 m for both instruments. Different radiosondes were used for the telemetry (RS-11G for SKYDEW and RS41 for CFH). The dew/frost point temperature from
SKYDEW agrees well with that from CFH below ~13 km. However, there is a difference reaching ~0.5 K in the stratosphere, with values that are sometimes slightly greater than the combined uncertainty ($k = 2$). Further investigation is needed to resolve this disagreement.

## 5.3 Comparison with radiosonde RH sensor

A chilled-mirror hygrometer is often used for evaluation of the RH sensor of radiosondes (e.g., Fujiwara et al., 2003; Vömel et al., 2007b; Nash et al., 2011). Ten soundings with Meisei RS-11G and SKYDEW were conducted on board the research vessel *Mirai* over the Northwestern Pacific Ocean from 27 May to 3 June 2021. These soundings were part of the Years of Maritime Continent Boreal Summer Monsoon 2021 campaign (YMC-BSM 2021; https://www.jamstec.go.jp/ymc/campaigns/IOP_YMC-BSM_2021.html). The Meisei RS-11G carries a capacitive sensor for
RH measurement (Kizu et al., 2018; Kobayashi et al., 2019). The RH can be calculated with the dew/frost point from SKYDEW and temperature from RS-11G using Eq. (3) or (4). Figure 14 shows the RH profiles from SKYDEW and RS-11G, and their differences on 27 May 2021. Figure 14 also shows the differences of eight profiles, omitting the two soundings that are contaminated by cloud. The differences above 15 km are probably due to the temperature humidity dependence (TUD) of the RS-11G sensor (Kizu et al., 2018). This bias will be improved for the RS-11G GDP version 2.






## 5.4 Comparison with Aura/MLS

The water vapor mixing ratio calculated from SKYDEW and the RS-11G pressure measurement was compared with the water vapor mixing ratio of the satellite Aura MLS version 4.23 dataset (Livesey et al., 2020) for validation of the stratospheric water vapor measurement. The Aura MLS measures water vapor above the upper troposphere with a vertical
resolution of 2–3 km. We compare seven profiles omitting the three soundings that are clearly contaminated from SKYDEW over the Pacific Ocean in May and June 2021. The SKYDEW frost point was vertically averaged over each 1 km. The mixing ratio from Aura MLS is not at exactly the same time and position as the SKYDEW soundings, and the water vapor mixing ratio is averaged over the range +/–2° latitude, +/–5° longitude. Figure 16 shows seven profiles of SKYDEW and Aura MLS. The mixing ratio value from SKYDEW increases above 25 km. This is probably because of contamination from
outgassing of the balloons and the flight trail (Hall et al., 2016). Below 25 km, the water vapor mixing ratio is in good agreement except for the profiles on 27 May and 1 and 3 June. For these three soundings, the mixing ratio above 20 km is higher than MLS by 1–1.5 ppmv. Figure 16 shows the ascent and descent profiles on 1 June; SKYDEW could measure the frost point below 23 km during descent. The descent measurement indicates a lower water vapor mixing ratio than the ascent measurement around 21–23 km. For the soundings on 1 and 3 June, the SKYDEWs passed through saturated air in the upper
troposphere. There is a possibility that the ascent measurements of these two profiles were affected by contamination from outgassing.

It is difficult to quantitatively estimate measurement bias due to contamination with only housekeeping data such as the scattered light intensity. If the descent measurement is successful, a comparison of the ascent and descent measurements is useful for removing the contaminated ascent measurement. In addition, the comparison with the satellite Aura/MLS is useful
for identifying contamination.

## 6 Summary

A Peltier-based chilled-mirror hygrometer, SKYDEW, has been developed to measure tropospheric and stratospheric water vapor. SKYDEW uses a PID controller as the feedback controller to maintain the equilibrium of the condensate on the
mirror. The behaviour of condensate on the mirror was observed by a microscope to investigate the characteristics and performance of this instrument. During the phase transition from water to ice on the mirror, it was found that the mirror temperature is equal to the dew point when the condensate on the mirror is mixed phase. We also observed that larger numbers of smaller ice crystals (~5 μm) are formed on the mirror at lower temperatures (–60 °C). Based on these results of condensate observation and the PID tuning in a chamber, the setting of the PID controller to maintain the condensate was
adjusted to retain slight oscillations of the scattered light signal from the mirror and mirror temperature so as to respond rapidly to dynamic change of atmospheric water vapor. These oscillations on the mirror temperature are smoothed with the



golden point method that selects the extrema of the scattered light intensity. More than 40 soundings with SKYDEW have been conducted since 2011 to improve and evaluate the measurement performance. The result of soundings at tropical and mid-latitudes demonstrated that SKYDEW is able to measure up to an altitude of 20–25 km for daytime soundings and

above 25 km for nighttime soundings, and the Peltier cooling creates a temperature difference over than 40 K in the stratosphere.

There are several sources of uncertainty: the stability of the PID controller and the error of the golden point determination, contamination error by cloud/rain droplets, the ambiguity of the condensate phase, aerosol effects and the curvature effect. The uncertainties from the golden point determination and filtering deviation error dominate, with values of

0.5 K for the case with large oscillations. These uncertainties are random and uncorrelated. The measurement uncertainty of mirror temperature derived from calibration is <0.1 K, and is a systematic error. The combined uncertainty for SKYDEW is <0.2 K during a stable measurement, but sometimes >0.5 K in the region where there are large oscillations of the mirror temperature.

Intercomparison soundings were conducted to evaluate the measurement uncertainty. Dual sounding with two

SKYDEWs showed agreement within their uncertainty and a small systematic bias. SKYDEW and CFH measurements almost agree within their uncertainties. However, a difference of up to ~0.5 K in the stratosphere was found. The comparisons with Aura MLS indicated good agreement when profiles contaminated by outgassing were excluded. SKYDEW descent data are also valuable for identifying contamination. For contamination-free soundings, implementation of the controlled descent such as done for NOAA FPH is a challenge for stratospheric water vapor measurements with

SKYDEW.

***Data availability.*** The sounding data of the YMC-BSM 2021 campaign are archived at the YMC Data Archive Center (https://www.jamstec.go.jp/ymc/campaigns/IOP_YMC-BSM_2021.html). The SKYDEW data in Tateno and Lindenberg will be archived at the data sever at GRUAN lead center after the SKYDEW data processing is certified by GRUAN. The RS-11G

GDP can be downloaded at GRUAN data server. (https://www.gruan.org/data/data-products/data-availability). Further data are available upon request.

***Author contributions.*** TS and KS designed the hardware and software of the SKYEW. MF and TS made the basic planning for SKYDEW development. TS analyzed the SKYDEW data and drafted the original manuscript. SO, JS and RD supported

the sounding to evaluate the SKYDEW performance. All authors reviewed and edited the manuscript.

***Competing interests.*** The authors declare that they have no conflict of interest.

***Acknowledgments.*** The author would like to thank the staffs of the Meisei Electric co., ltd for their technical support. The

author would like to appreciate Mr. M. Ibata and Mr. Y. Kanai for providing useful comments and support. The author



thanks the member of the Soundings of Ozone and Water in the Equatorial Region (SOWER) project, the members of Lembaga Penerbangan dan Antariksa Nasional (LAPAN) of Indonesia, and the staff of the Lindenberg observatory and Tateno Aerological Observatory for their collaboration at the soundings. The authors would like to thank Peter Oelsner at the Lindenberg observatory for creating the decoding software for SKYDEW XDATA version. The early phase of this study

was in part supported by JSPS KAKENHI Grant Numbers JP22740306, and by the Institute of Space and Astronautical Science/Japan Aero-space Exploration Agency (ISAS/JAXA).

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



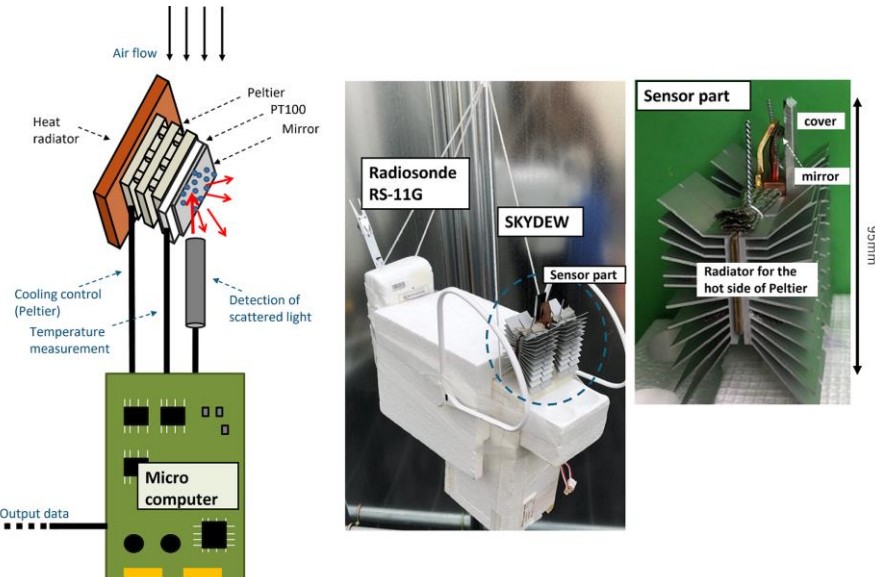

**Figure 1:** Left: Schematic of SKYDEW. Right: Photograph of SKYDEW with the Meisei RS-11G and a close-up of the sensor part. The sensor probe is positioned on top because SKYDEW is designed for measurement during balloon ascent.



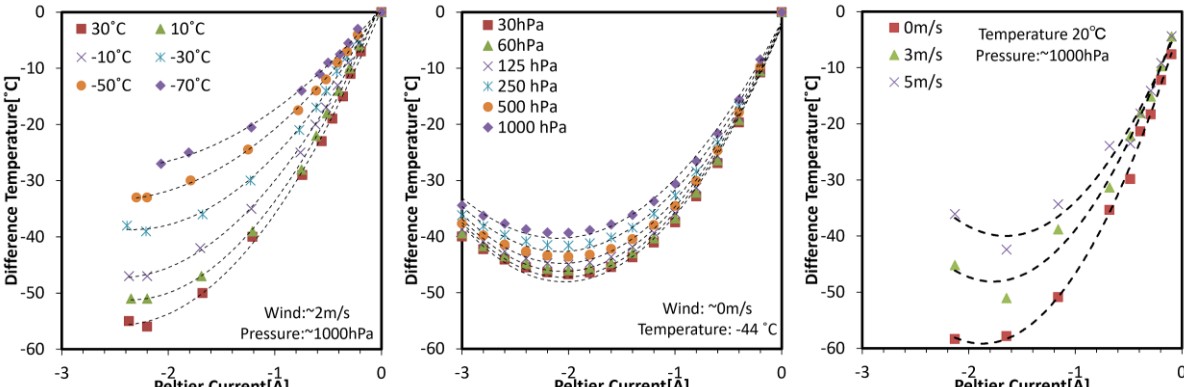


**Figure 2:** Cooling capability of the Peltier device used in this study. Shown are the temperature differences between the cold side of the Peltier device and the ambient air versus the Peltier current under various conditions obtained in a thermostatic chamber. The panels show the dependence on (left) air temperature, (center) air pressure, and (right) airflow rate.




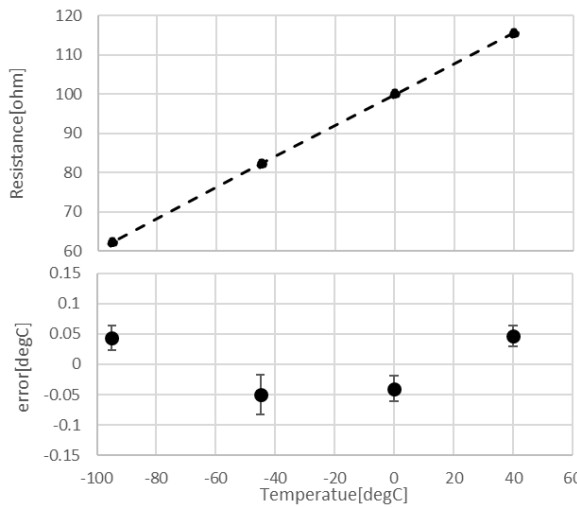

**Figure 3:** Top panel: relationship between the resistance of the PT100 and the reference temperature. Bottom panel: errors from the calibration curve. The error bar indicates the standard deviation for 20 sensors.


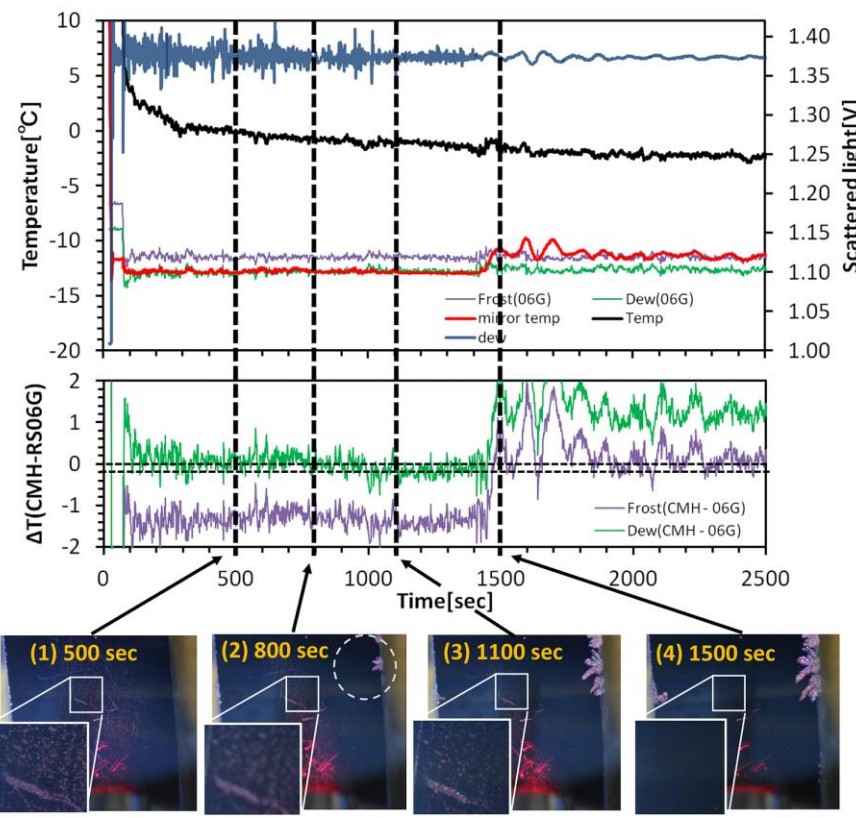

**Figure 4:** Top panel: time series of the mirror temperature (red) and dew/frost point temperature (purple and green) calculated from the Meisei RS-06G air temperature (black) and RH in the laboratory and the scattered light signal (blue) which corresponds a right axis. The RH from RS-06G is corrected by a constant offset of 2.5% RH for this experiment to
agree with the SKYDEW RH. Bottom panel: differences between the mirror temperature and the frost point temperature (purple line) and dew point temperature (green line). Photographs shows the condensate on the mirror. The vertical dashed lines indicate the time when the photos were taken.





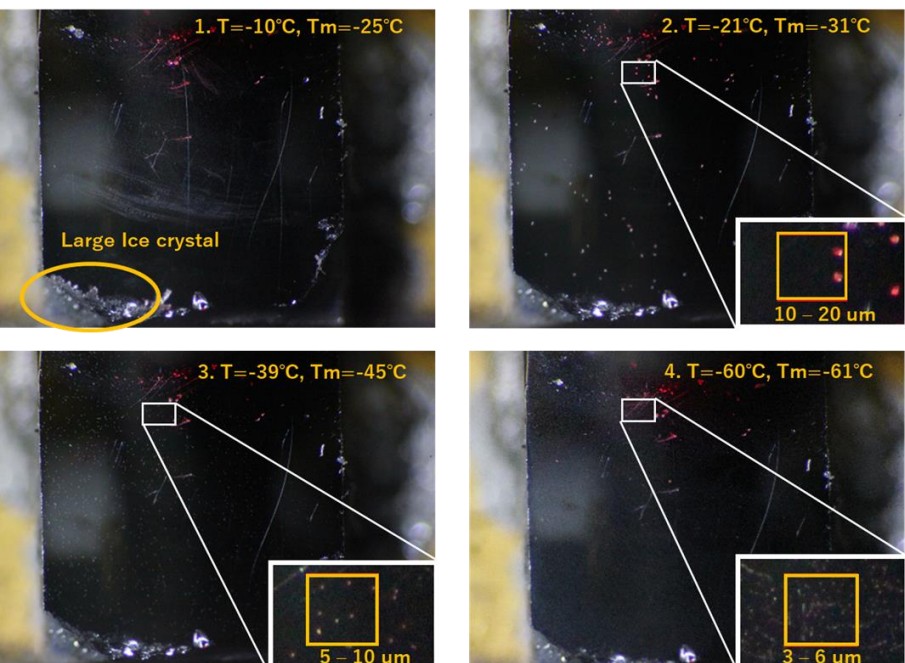

**Figure 5:** Condensates on the mirror at temperatures of –10 °C (top left), –20 °C (top right), –40 °C (bottom left), and –

60 °C (bottom right) in the thermostatic chamber.




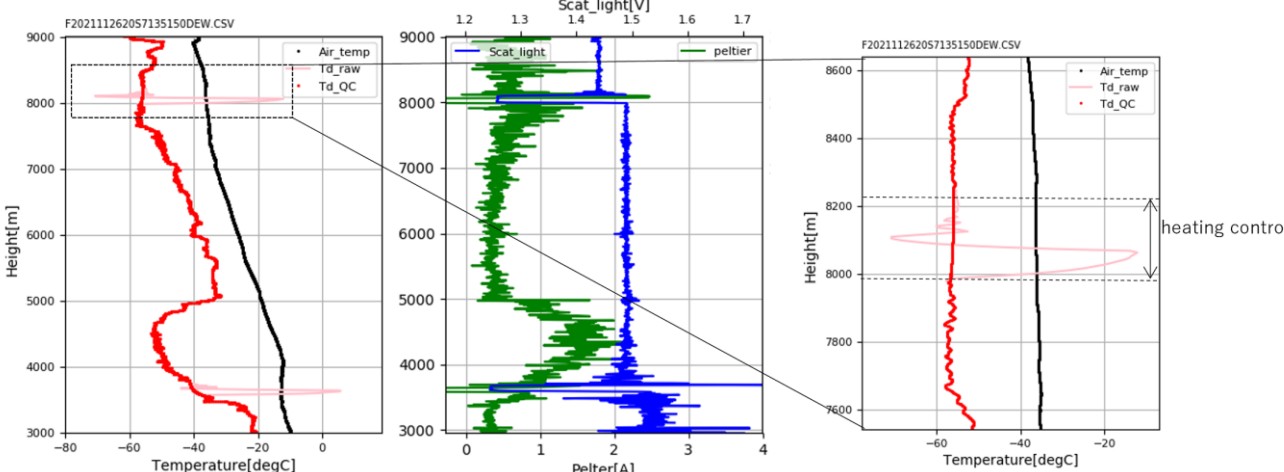

**Figure 6:** Heating control stage at –36 °C and –58 °C for the profile on 26 November 2021 at Tateno, Japan (36.06° N, 140.13° E). For each control, the mirror is heated for ~5 s to clear the mirror surface. When the mirror temperature (pink) is heated, the scattered light signal (blue) goes down to the base level, indicating that the frost layer disappears. Red lines indicate the mirror temperatures after quality control and interpolation.




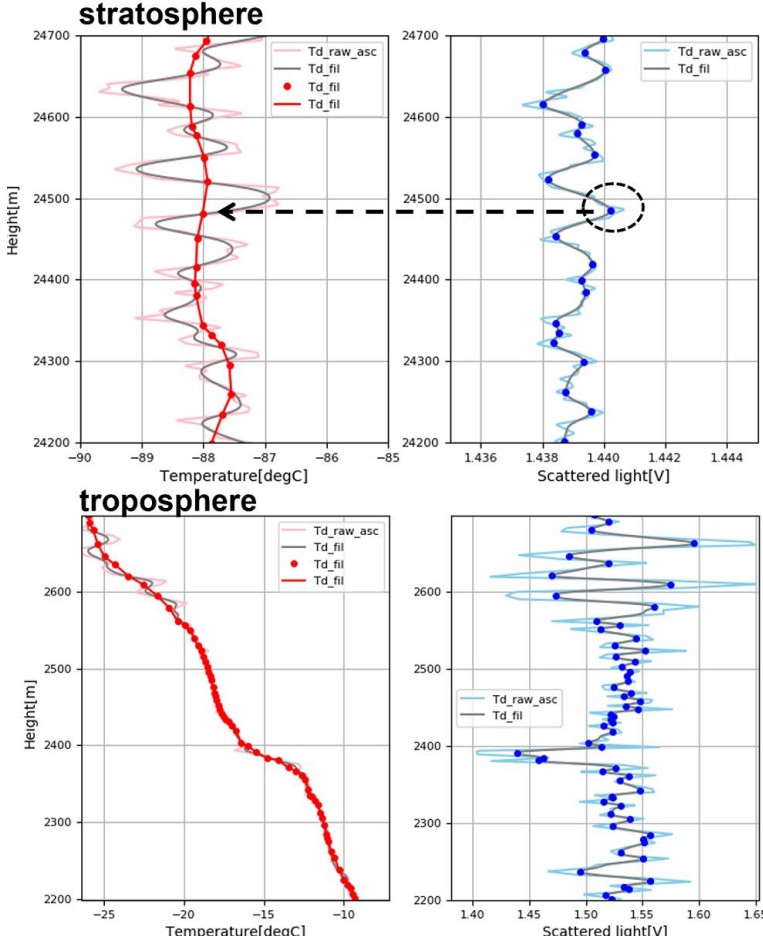

**Figure 7:** Golden point detection for the sounding shown in Fig. 6. Example profiles are shown of dew/frost point in the troposphere (bottom) and stratosphere (top). The right panels show the scattered light, including the raw data (light blue) and its smoothed profile (grey line). The blue dots are the detected gold points. The left panel shows the mirror temperature, including the raw data (pink) and its smoothed profile (grey line). Red dots are mirror temperatures corresponding to golden points, which are frost points.



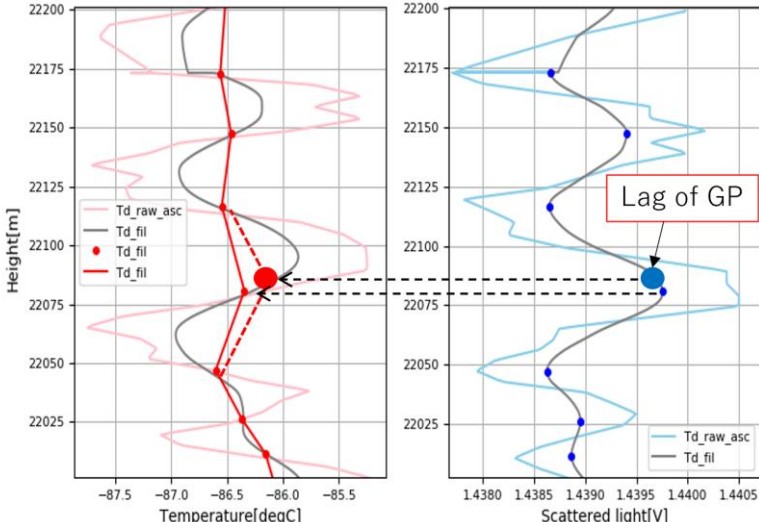

**Figure 8.** As for Fig. 7, but explaining the timing error of the golden point. If the equilibrium point of scattered light is mistakenly detected as the golden point (large blue dot in the right panel), the frost point is erroneously determined (large red dot in the left panel).



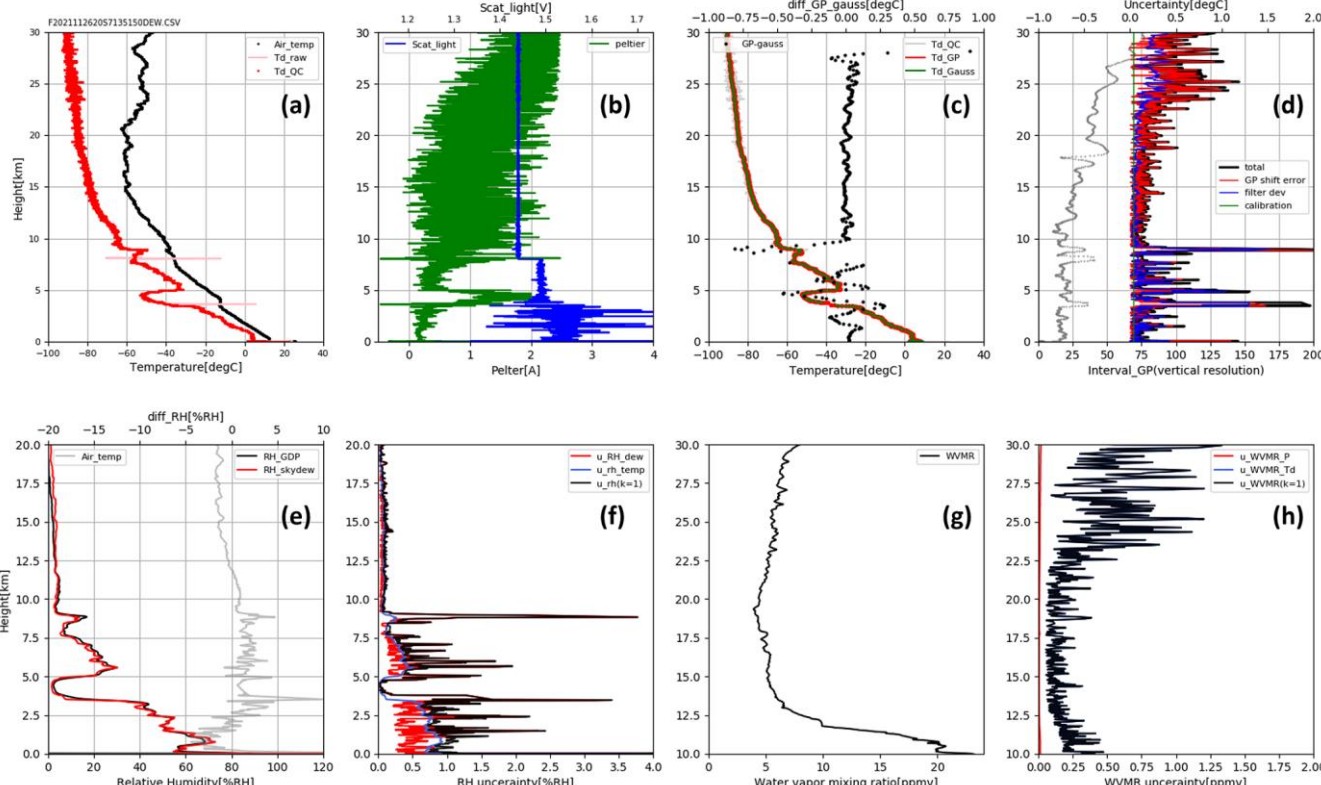

**Figure 9.** Profiles of dew/frost point and its uncertainty from SKYDEW on 26 November 2021 at Tateno. Mirror temperature and air temperature (a), scattered light intensity and Peltier current (b), smoothed dew/frost point with the golden point method and gaussian filter (c), uncertainty ($k = 1$) and the vertical resolution (d), RH profiles and its difference (e), uncertainty of RH calculated from SKYDEW ($k = 1$) (f), water vapor mixing ratio from SKYDEW (g), and uncertainty of water vapor mixing ratio($k = 1$) (h) .



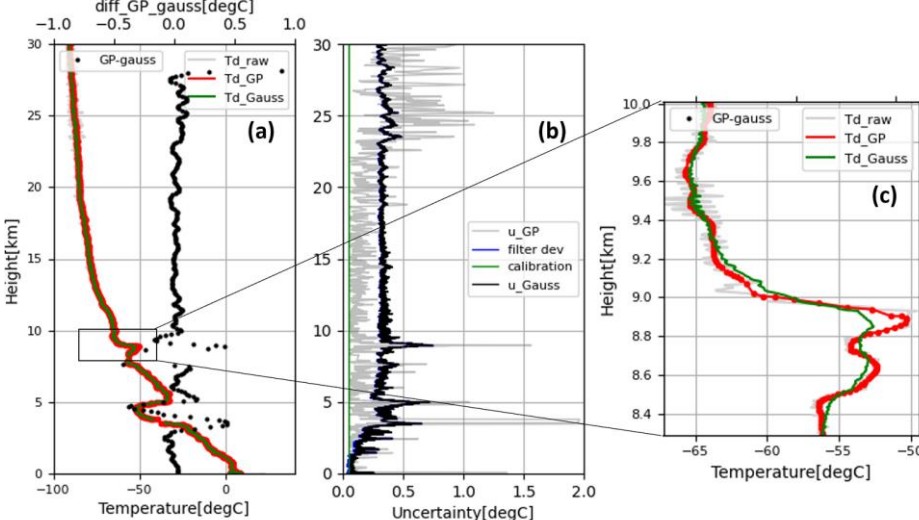

**Figure 10:** Frost point profiles on 26 November 2021 at Tateno smoothed with the golden point method (red line) and gaussian filter with σ = 13 (green line). (a) Whole profile of frost point and black dots indicates differences between frost points smoothed with each method. (b) Frost point uncertainty in the case of a gaussian filter, filtering deviation (blue), calibration error (green), and total uncertainty (black). Gray line is the uncertainty of the golden point method. (c) Close-up profiles between 8.4 and 10 km. The grey line indicates the raw profile of mirror temperature.



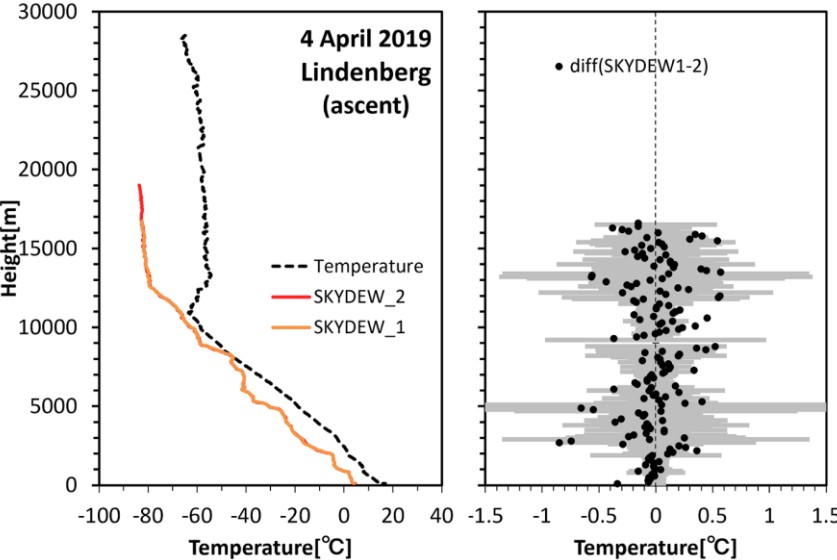

**Figure 11.** Dual sounding with two SKYDEWs on 4 April 2019 in Lindenberg, Germany (52.21° N, 14.12° E). The left panel shows the dew/frost point profile (red and orange lines) and air temperature measured by radiosonde (black line). In the right panel, the black dots are the differences between the two SKYDEW measurements and light grey indicates the combined uncertainties ($k = 2$).





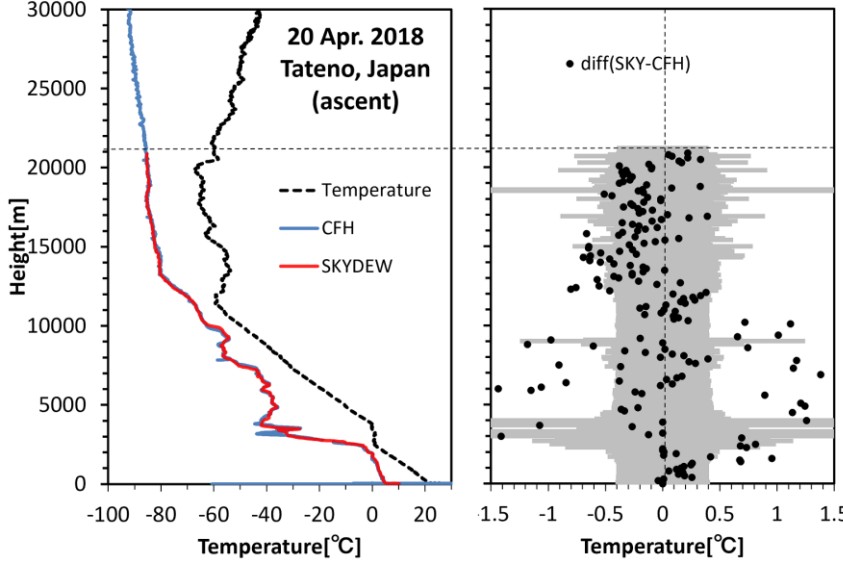


**Figure 12.** Comparison of soundings with CFH (blue line) and SKYDEW (red line) at Tateno on 20 April 2018. The cooling limit for SKYDEW was 21 km for this sounding, as shown by the black dashed line. In the right panel, the black dots are the differences and light grey indicates the combined uncertainties ($k = 2$) calculated from SKYDEW and CFH measurement uncertainties. The uncertainty of CFH was assumed as a constant value of 0.2 K.




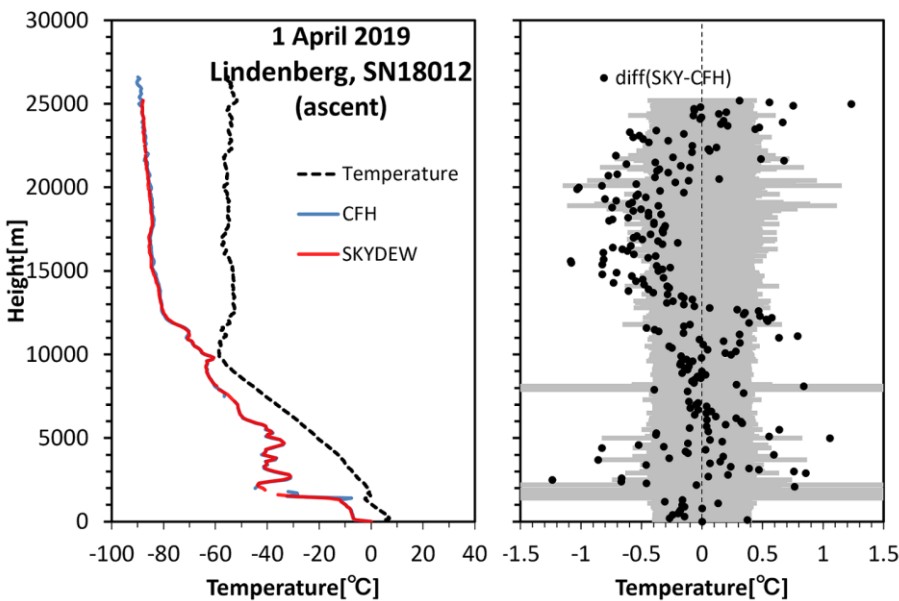

**Figure 13.** As in Fig. 12, but for the profile at Lindenberg on 1 April 2019.






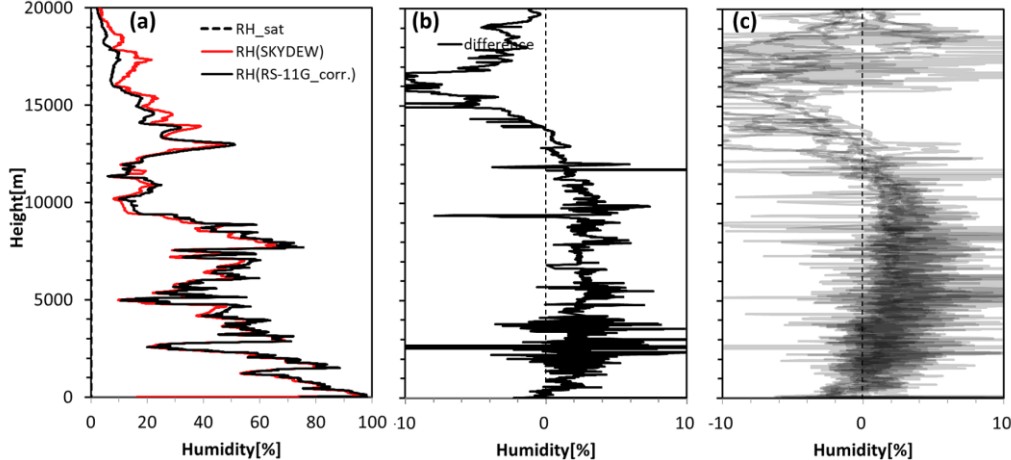

**Figure 14.** Comparison of RH with SKYDEW and RS-11G. The left panel shows the RH profiles on 27 May 2021 with SKYDEW (red) and RS-11G (black). Black dashed line indicates the RH of ice saturation. The centre panel shows the difference. The right panel shows the differences of eight soundings over the Pacific Ocean between 27 May and 3 June.




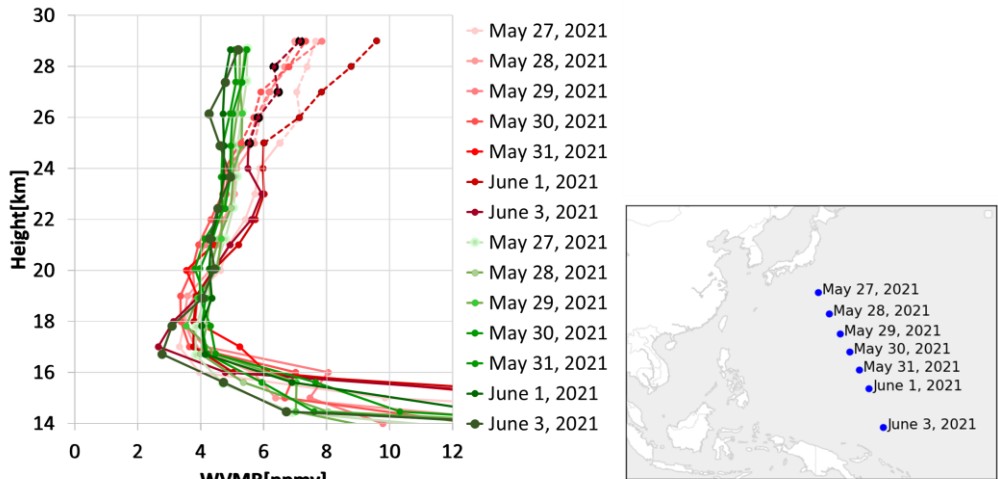

**Figure 15.** Water vapor mixing ratio from SKYDEW (red) and Aura MLS (green). Dashed lines indicate the suspected contaminated profiles. Right panel shows the launching points and dates for SKYDEW.



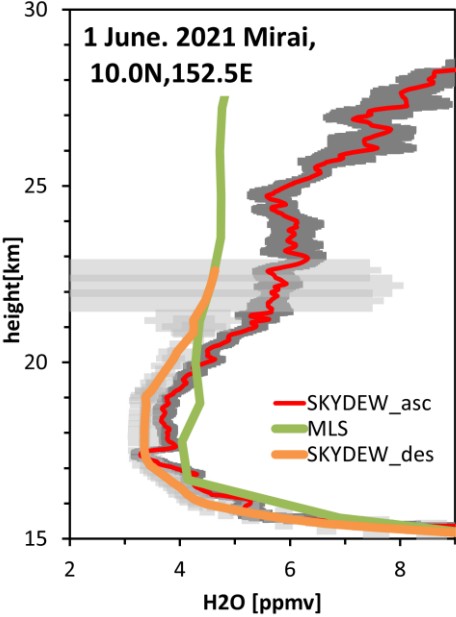

**Figure 16.** Water vapor mixing ratio from SKYDEW and MLS on 1 June 2021. Red line is the ascent profile and orange line the descent profile. Gray shading indicates the uncertainty ($k = 2$).





**Table 1:** Uncertainty sources and estimates for SKYDEW.

| Uncertainty parameter | Value (k=1) | (Un)/correlated |
|---|---|---|
| Mirror_temp measurement, u_mirror | $0.052 = (u\_mrr1^2 + u\_mrr2^2 + u\_mrr3^2)^{0.5}$ | correlated |
|   PT temperature calibration, u_mrr1 | $0.07/\sqrt{3}$ | correlated |
|   Resistance measurement u_mrr2 | $0.005\sqrt{3}$ | correlated |
|   Thermal gradient of mirror, u_mrr3 | $0.015/\sqrt{3}$ | correlated |
| Golden point detection error, u_GP_error | ~0.6 at max | uncorrelated |
| Filtering deviation, u_filter | ~1.0K at max | uncorrelated |
| Contamination by cloud | Depend on the situation strongly. | correlated |
| Aerosol effect and curvature effect | negligible | correlated |
| Total uncertainty, u_DP | $= (u\_mirror^2 + u\_GP\_error^2 + u\_filter^2)^{0.5}$ | |
