# Peer review of "Development of a Peltier-based chilled-mirror hygrometer SKYDEW for tropospheric and lower stratospheric water vapor measurements"

_EGUsphere, 2024_

## Author Comment (AC1)

**Reply to comments by Anonymous Referee #1**

The authors appreciate for your constructive comments and suggestions. We will revise the manuscript by taking your comments into account.

Below are the *reviewer #1 comments* in blue and our response in black.

*Summary:*

*The manuscript by Sugidachi et al. describes a new Peltier cooled frost-point hygrometer to measure stratospheric and tropospheric water vapor. This development expands to ability to measure stratospheric water vapor, which is desperately needed.*

*The manuscript describes some instrumental details as well as some comparisons and uncertainty estimates. It is heavily based on previous work on cryogenically cooled frost-point hygrometers and adopts many of the same ideas and principles.*

*The descriptions are often a little superficial and skip on several details. There appear to be many more profiles than the authors show here, which would allow them to do a statistical comparison between Skydew and CFH or Skydew and MLS. This seems to be missing and would add significantly to support the impact that this instrument may have. As is, the selection of profiles appears a little selective.*

*Below, I outline several points for improvement of the manuscript. I assume that these will take some time to implement. For these reasons I am hesitant to recommend publication of the manuscript without some major revisions.*

Please find below our responses to your comments. In most cases, we agree with your suggestions and will revise the manuscript accordingly.

*Detailed comments:*

*In the description of the sensor the authors state that a temperature difference of 90°C can be achieved. Here they should clarify that this is the temperature difference between hot and cold side, not between ambient temperature and cold side. Later, they state, that the achievable temperature difference between ambient temperature and cold side can be > 55°C at the surface and 30°C at an ambient temperature of -70°C. In the summary they state a temperature difference of 40 K in the stratosphere. Figure 2 clarifies that these values only apply at a wind*

*speed of 0 m/s. Adding wind of 5 m/s, a realistic temperature difference may probably by around 20°C smaller. Figure 12 seems to indicate a cooling limit of 25 K. In line 365, the authors state that an extreme limit at which Skydew cannot take measurements would be dew-point depressions larger than 60°C at the surface. Considering the statements above, that limit seems to be more than optimistic under real world flight conditions. It would be good to expand that discussion and consolidate the different statements throughout the manuscript.*

The maximum cooling capacity of the Peltier element used for SKYDEW is a temperature difference $\Delta T$ of 90°C as described in the datasheet. But this value is measured under ideal conditions, namely in a vacuum and at 27°C. We acknowledge that this sentence may cause confusion, so we will delete the sentences at lines 136 and 137.

The left panel of Figure 2 displays the temperature difference at a wind speed of 2 m/s, not 0 m/s. Indeed, the $\Delta T$ of >55 K mentioned at line 226 was measured at a wind speed of 2 m/s, not 5 m/s which is the standard ascent rate. We lack the experimental facility to generate a wind speed of 5 m/s under low temperature and low pressure conditions. Therefore, we estimated the cooling limit using theoretical Equation (7). We will explicitly describe this fact to reflect the temperature difference $\Delta T$ at an ascent rate of 5 m/s throughout the manuscript. (The estimated $\Delta T$ is approximately 45 K at 20°C and 1000 hPa, and $\Delta T$ is 30 K at -70°C and 100 hPa in the lower stratosphere under 5 m/s ascent rate.)

We will add estimated values using Equations (7) and (8) as dashed lines for each condition in Figure 2. Furthermore, the values for the lower stratosphere (-70°C, 100hPa, and 5m/s ascent) will be represented as a grey line. In the original manuscript, the dashed line in Figure 2 was simply a fitting curve for each condition.

Figure 12 shows a sounding during the daytime. Therefore, because the hot side of the Peltier element is heated by solar radiation, the cooling limit is lower than the estimated values. We will provide more detailed information on this below.

We will remove the description "(dewpoint depression > 60 K at the surface)" at line 365. In this section, we explain the quality control, and the temperature difference is not used in the actual QC procedure. We use only the Peltier current and scattered light to detect the cooling limit.

[Figure]

Figure R1-1. The revised figure 2 about the cooling performance. The dashed lines indicate the theoretical values calculated by Eq. (7), using the parameter $\alpha = 0.0099, \beta = 0.0432, R_e = 0.48$, and $S = 0.00025$. See the detailed information shown within each panel for different symbols, colors, and lines.

*Related to this, the upper altitude at which the Skydew sensor can measure is not well supported. The authors point out that there is a systematic difference between day and nighttime measurements. Is the 25 km limit (not considering contamination) for nighttime achievable for all geographic locations? Is this limit a maximum under optimal conditions, or an average? Is this an optimistic estimate?*

The limit of 25 km at nighttime is deemed achievable in many locations, including tropical and mid-latitude regions. However, the performance in polar regions remains uncertain due to the limited soundings conducted with SKYDEW in these areas.

The estimated cooling limits are plotted on the actual sounding results in the Figure R1-2. The top panel shows the nighttime sounding at mid-latitude. The estimated maximum cooling, represented by the grey line in the left panel, is calculated using the actual hot side temperature and the temperature difference estimated by Equation (7). For this example, the estimated maximum cooling is consistently lower than the dewpoint depression below 25km, which is within the cooling limit. On the other hand, the mirror temperature (frost point) reaches the estimated cooling limit at 25 km. This suggests that the cooling limit for this sounding is around 25 km. However, the mirror temperature is lower than the estimated cooling limit above 25km. As a result, this estimation might be conservative, and there is a possibility that the SKYDEW could cool slightly more in reality.

The bottom panel in Figure R1-2 shows the nighttime sounding for the tropics. The estimated ΔT at the tropopause is less than 30 K due to the low temperature. However, since the dewpoint depression is typically small at this altitude, SKYDEW does not face the cooling limit even above the tropopause. At higher altitudes, the dewpoint depression is larger, and SKYDEW

needs to create a larger temperature difference to maintain the mirror temperature at the frost point. The maximum cooling is also larger due to the low air density, which results in less heat transfer. As a result, the cooling limit is located near 30 km for this sounding. During the daytime, the hot side is heated by solar radiation, especially in the stratosphere. This leads to a lower cooling limit.

[Figure]

Figure R1-2. Vertical profiles of the comparison of observed temperature differences and the cooling limit estimated by Equation (7) for nighttime measurements. The top panel shows the profiles on 26 November 2021 at Tateno, Japan (36.06° N, 140.13° E). The bottom panel shows the profiles on 3 June 2021 on board the research vessel Mirai over the Northwestern Pacific Ocean (1.4° N, 155.5° E)

*In the abstract, the authors report a difference of 0.5 K between the CFH and Skydew. This does not properly summarize the results in the text and gives a wrong impression of the comparison between the two instruments. This difference was found in only one of two soundings they discuss. There are more comparisons between the two instruments, which are not discussed in*

*There are over 40 soundings using Skydew with different development stages. The descriptions in this paper, I assume, refer mostly to the current two-stage Peltier cooled version. How many soundings have there been using that version compared to previous ethanol cooled versions? How many comparisons with CFH or other reference quality instruments are there? Can the authors provide statistically significant results? How robust is the instrument performance across many soundings, considering that these instruments are inherently disposable?*

So far, we have compared the final production version of SKYDEW with the CFH over ten times (Sugidachi 2019). The profiles from the five intercomparison soundings, excluding soundings with some doubtful data either of the two instruments, indicate that the SKYDEW readings are slightly smaller (<~0.5K) compared to the CFH in the lower stratosphere. However, these data have not yet been analyzed using the golden point method. We plan to re-analyze these data using a new algorithm, modified in response to the reviewer's comments, and will present the results in the revised manuscript.

In terms of the comparison between the previous ethanol-cooled version and the final version, we have not conducted a comparison between these two, though we do not expect essential differences because the only differences are the heatsink shape and the use of ethanol.

More than ten dual soundings using two SKYDEW instruments have been performed at Lindenberg since 2023. However, as these results will be reported in another paper, we will not include these dual sounding results in this paper.

References:

Sugidachi, T. : Meisei SKYDEW instrument: analysis of results from Lindenberg campaign, in 11th GRUAN Implementation and Coordination Meeting, Singapore, 23 May 2019 https://www.gruan.org/gruan/editor/documents/meetings/icm-11/pres/pres_0803_Sugidachi_SKYDEW.pdf (last access: 2 June 2024)

*Line 91: The data quality of the cryogenically cooled instruments has certainly changed over the last 4 decades. The authors could just delete that statement.*
We will delete the sentence of line 91.

*The calibration uncertainty will be the limiting uncertainty for long term trends. The calibration uncertainty is quite reasonable, but can the authors show how they verify this uncertainty in long term production.*

Each PT100 sensor of SKYDEW is calibrated against an SI-traceable reference. We will add a

description about the SI-traceability in the manuscript.

"Each PT100 is calibrated at four temperatures (40°C, 0°C, −45°C, and −95°C) against an SI-traceable reference to characterize the individual Resistance-Temperature (R–T) relationship of the PT100."

*Does the instrument measure the temperature of the warm side of the Peltier device to verify that the sensor performs within the expected range? If so, if would be very useful to show these results.*

Yes, we consistently monitor the temperature of the hot side of the Peltier element. We will add this temperature profile to Figures 9, 12 and 13 in the revised manuscript.

*Equation 7 is not obvious. If the heat conductivity across the device is considered small and the heat input into the cold side is small, then the temperature delta across the device becomes proportional to the resistive heating of the device. That does not appear to make sense for a Peltier device. The source for equation 7 is cited as Sugidachi (2014). The source for the same equation in that thesis is cited as Sugidachi (2011). The link in that reference no longer works and this thesis cannot be easily accessed. Thus, it is difficult to trace the reasoning for this equation. (On that note, in other places as well, the authors do not cite the original literature, but rather papers citing earlier publications. It would be better if they directly cited the earlier or original work.)*

The explanation in the original manuscript was indeed insufficient. We will add the description of the processes to derive Equation (7) in the revised manuscript as below:

The heat balance per unit time, which includes the heat transfer between the ambient air, is written as follows:

$$m\,c\,\frac{dT_m}{dt} = \alpha\,T_m\,I + \frac{R_e I^2}{2} + \beta(T_h - T_m) + HS(T_{\text{air}} - T_m)$$

(1)

where $T_m$ is the temperature of the mirror surface at time t , $m$ is the mass of the mirror and the Peltier element, $c$ is their specific heat, $R_e$ [ohms] is the resistance of the Peltier device, $I$ [A] is the current through the Peltier device, $\alpha$ [V K–1] is the coefficient derived from the material and the size of the Peltier device, $T_{\text{air}}$ is air temperature, ß [W K⁻¹] is the coefficient associated with the thermal conductivity of the Peltier device, $H$ [W m⁻² K⁻¹] is the heat transfer coefficient between the mirror surface and air temperature, and $H$ [m2] is the mirror surface area. Assuming that the heat sink quickly equalizes with the surrounding air and can always be approximated as $T_h = T_a$, $T_m$ can be expressed as follows:

$$T_m = \Delta T\left\{1 - exp\left(-\frac{-\alpha I + \beta + HS}{m\,c}\right)t\right\} + T_a,$$

$$\Delta T = \frac{\frac{RI^2}{2} + \alpha I T_a}{-\alpha I + \beta + HS}.$$

(2)

In this equation, at t=0, $T_m = T_a$, indicates that the mirror temperature is equal to the ambient temperature. Over time, the mirror's temperature decreases exponentially, implying that it creates the maximum temperature difference $\Delta T$ after sufficient time has passed. The sign of the Peltier current is by (our) definition negative in cooling mode and positive in heating mode. The reference, Sugidachi (2011), has now been made publicly accessible. We will add the URL to the reference list in the manuscript. This will make it easier for readers to access the source.

Sugidachi, T.: Development of a balloon-borne hygrometer for climate monitoring, master's thesis at Graduate school of Environmental Science, Hokkaido University, 138 pp. (*in Japanese*), https://eprints.lib.hokudai.ac.jp/dspace/bitstream/2115/92715/3/takuji_sugidachi_2011.pdf (last access: 10 July 2024), 2011

*Section 2.6.1: During the time of the phase transition, the instrument is most likely controlling not around the dew-point or frost-point, but rather around the change in reflectivity between the two phases. During this period, the dew-point or frost-point is not defined by the mirror temperature. However, the control around the dew-point seems to stable until about 1400 s, i.e., super cooled water persists for that long (not 500 s). A region of ice crystals seems to grow in the upper right corner of the mirror. Does the detector even see these very large crystals? Do the authors have any information about the temperature gradients across the mirror? Could the crystal growth in the corner be caused by a significant temperature gradient? (Same questions for the very large ice crystals at the edge of the mirror in Figure 5.)*

Indeed, when both ice and water co-exist on the mirror, the mirror temperature is not strictly the dew point nor frost point. However, this experiment shows that the mirror temperature is close to the dew point as long as supercooled water remains on the mirror. We will add the phase transition timing on the profile of dew/frost point in the revised manuscript to clarify the additional uncertainty due to phase ambiguity near this altitude.

We have not measured the temperature gradients on the mirror. Instead, we have roughly estimated the temperature gradient in the thickness direction of the mirror using Fourier's law. The mirror material is silicon, which has high thermal conductivity (148 W/mK). Because the mirror is located on the Peltier element, the top side and bottom side of the mirror have a slight temperature difference due to the internal temperature gradient. For instance, if the cold side of the Peltier element is cooled down 30K compared to the ambient air, and assuming $H$ =50, the thickness of the mirror $L$ =0.28mm, the temperature difference is 0.015K. This potential error

is included in the uncertainty budget.

The reason for the large ice growth on the edge of the mirror is considered to be because the mirror edge acts as ice condensation nuclei. The polished surface of silicon has little roughness, but the sharp edge is easily ice nucleated. The SKYDEW is sensitive to this large ice on the edge as well. This large ice is not preferable because it causes a slow response due to the slow evaporation/growth rate.

*Lines 307 ff: There is one fundamental difference between cloud physics and the condensation processes on the mirror: The mirror itself can act as an ice-forming nuclei. Therefore, the temperature dependence is likely very different than that for the free atmosphere. The growth rates are possibly similar.*

We will delete the Line 306-309.

*Sections 3.2 and 4.2: The derivation of the golden points and their uncertainty is very unclear. As written, the authors first create 5 Hz data by linear interpolation of the original 1 Hz data, then electrical noise is removed, lastly the data are smoothed using a Gaussian filter with a width of 1 to 3 s depending on altitude. The authors should remove the electrical noise first. How large is the electrical noise and how can it be distinguished from that generated by the PID controller? Secondly, why do they create 5 Hz data, if they are smoothed anyway? The linear interpolation combined with the Gaussian smoothing may even generate a little bit of extra noise, although that may be small. In the uncertainty estimation they use the number of 5 Hz data points in their statistics and assume they are random and uncorrelated. They are clearly not random and uncorrelated since they were created by linear interpolation and smoothing. This needs to be corrected. How is the uncertainty of the mirror temperature in the golden point selection process quantified? Is it the uncertainty of ±1 s in the detection of the golden point and the spread of the mirror temperature over these three data points? Figure 8 does not clarify this question. In line 383 the authors state that the extracted points are smoothed with a Gaussian filter. However, the extracted points are unevenly spaced, and Gaussian smoothing would achieve odd results under these conditions.*

Regarding the removal of electrical noise, we found an error in the parameter setting for the width of the Gaussian filter in the original manuscript. The profiles in Figure 7 were generated using a Gaussian filter with a width of $0.4 - 1.5$ seconds. During the development phase, we adjusted the width to be between $0 - 1$ second at the surface and about $1 - 3$ seconds in the stratosphere. In the figure in the original manuscript, these parameters are mixed. We will

correct this in the revised manuscript.

When there is water droplet on the mirror, the fluctuation of scattered light signal is strong enough to overcome the noise. Therefore, there is no problem with using a narrow-width Gaussian filter. On the other hand, the fluctuation of scattered light signal is weak when there is ice particle on the mirror and can cause detection errors for the golden point if smoothing is not applied. Figure R1-3 shows an example of golden point detection with and without a Gaussian filter. For the troposphere, it is difficult to clearly distinguish between noise and real signal. However, the signal-to-noise ratio is good. The effect of the Gaussian filter is negligible and it is considered to be almost not necessary. For the stratosphere, there are many detection errors of golden point because the maximum and minimum points generated by the noise are detected in cases without smoothing. The oscillation period caused by the PID controller is typically 10 - 20 seconds. To avoid deleting these oscillations, we applied a Gaussian filter with a width of less than 3 seconds.

Regarding the creation of 5Hz data by linear interpolation, we assume that the golden point may sometime exist between the 1Hz sampling intervals. Figure R1-4 shows an example of detection from 1Hz data and from 5Hz data. The asterisks indicate the golden point from the smoothed 1Hz sampling data. This detection has a small shift from the detection point from 5Hz data. However, this shift does not cause a large error in frost point determination, and this value is negligible compared to the estimated uncertainty derived from the golden point error. Therefore, we decided not to apply up sampling with linear interpolation. Figure R1-5 shows the case of 1Hz data for the original Figure 9. There does not seem to be a significant difference. Increasing the original sampling rate could improve golden point detection. Although this cannot be implemented immediately, we will consider implementing higher-temporal-resolution sampling for the next firmware version in the future.

We described that the uncertainty of this golden point detection is random. As you pointed out, this uncertainty is correlated vertically because of smoothing. But this uncertainty should be treated as uncorrelated among different sounding profiles. We will modify lines 419-420.

Regarding the uncertainty estimation of golden point detection, Figure 8 will be revised as shown in Figure R1-6. Grey shadow area indicates the area between the error of +/-1 second. The absolute values of +1 s and -1 s error are not equal. So, the larger values are used for the uncertainty estimation as follows:

$$u_{GPerror} = \frac{|Tm_{GP} - Tm_{GP+1}|}{\sqrt{3}}, \text{ for } |Tm_{GP} - Tm_{GP+1}| > |Tm_{GP} - Tm_{GP-1}|$$

$$u_{GPerror} = \frac{|Tm_{GP} - Tm_{GP-1}|}{\sqrt{3}}, \text{ for } |Tm_{GP} - Tm_{GP+1}| < |Tm_{GP} - Tm_{GP-1}|$$

(3)

Regarding the smoothing after the selection of golden point, we used the Gaussian filter on unevenly spaced data. This process will be modified in the revised manuscript. We will generate

1 Hz data with resampling before applying the Gaussian filter.

As described above, we will make significant modifications to the process and related figures. These modifications, including the revised algorithm, figures, and detailed explanations, will be included in the revised manuscript.

[Figure]

Figure R1-3 Example profiles of golden point detection with and without a Gaussian filter. The left panel displays the profile with a Gaussian filter, which is the same as Figure 7 in the original manuscript. The right panel shows the profiles without a Gaussian filter. The bottom panels represent the profile at the troposphere, while the top panels represent the profile at the stratosphere.

[Figure]

Figure R1-4. The difference because of the detection from 1 Hz and 5 Hz data. Left panel is the mirror temperature, and light red line is raw data, grey line is smoothed one, dot is the golden point detected from the 5Hz data, and asterisk is the golden point detected from the 1Hz data, right panel is the scattered light signal. Dot and asterisk is same as mirror temperature.

[Figure]

Figure R1-5. The profile of golden points detected from 1Hz data without up-sampling. The top panels are the same as panels (c) and (d) in Figure 9 of the original manuscript. The bottom panels display the same content, but the profile is derived from 1Hz data.

[Figure]

Figure R1-6. The profiles explaining the detection timing error of the golden point. The asterisks indicate the positions with a shift of +/- 1 second. The grey shadow represent the areas surrounding the error golden points.

*Related to the derivation of the golden points and the smoothing, the authors should make a statement how that affects the vertical resolution of the observations and how the vertical resolution is given in the processed data.*

We have described the topic of vertical resolution at lines 384-387 in the original manuscript. In the revised manuscript, we will replace the term "vertical interval" with "vertical resolution" for clarity.

*The authors also state (line 548 ff) that the aggressive PID tuning and larger oscillations are required for the golden point detection. However, there is no such requirement in the derivation or golden points. The method should work with any amplitude of oscillation since it only requires a short period of constant reflectivity.*

Yes. We will modify this sentence.
"Smaller oscillation is required for better measurement, although the oscillation with short periods due to the aggressive setting of the PID controller is needed to detect the golden point with high vertical resolution."

*The authors find that direct gaussian smoothing of the data leads to slightly lower frost-point temperatures than that derived from the golden point method.*

*Section 5.2: There are more comparisons between the two instruments, which should be shown and discussed. Is the difference seen in the Lindenberg profile also seen in other comparisons? If so, this should be reported. If not, this should be reported as well. If any of the instruments in the other comparisons did not provide data in any of these additional comparisons, this could also be reported. As is, showing just two comparisons seems a little selective.*

Yes. We agree with your suggestion and will include additional comparisons between the two instruments in the revised manuscript.

*Section 5.4: Looking at the data referenced by the authors, the comparison with MLS in the tropopause region seems to look a little better than shown by the authors. The averaging kernel of MLS is broader than 1 km. The Skydew data should be smoothed using the MLS averaging kernel, or at least smoothed to the same vertical resolution. They may need to review how they do the comparison. However, as a result the disagreement on ascent starts at a lower altitude. It would be helpful, if the authors could create a relative difference plot with MLS as reference.*

We will try to compare with MLS data by applying the MLS averaging kernels to the Skydew data. We will also show the difference plot in the revised manuscript.

*One of the three "contaminated" profiles shows a significant jump in the upper troposphere, which is unlikely contamination. This may indicate additional complications not described by the authors.*

If you are referring to the sounding on May 27, it's possible that cloud particles may have influenced the scattered light signal at 13 km. The oscillation in the scattered light signal at approximately 13 km corresponds to the near-saturated layer shown in the frost point profile. When SKYDEW passes through a cloud layer, such oscillation often occurs, especially in the upper troposphere, due to aggressive PID tuning. The sounding on June 1 was similar to that on June 3, with a thick cloud layer present in the upper troposphere.
We will clarify this point in the revised manuscript.

[Figure]

Figure R1-6. The sounding profile on 27 May. It appears that the SKYDEW detected something at 13 km. This could indeed be cloud particles.

*Lines 628 ff: The sensor sits on top of Styrofoam box, which is very good for ascent measurements, but very bad for descent measurements. How would descent measurements be better for identifying contamination?*

We will indeed remove the sentence: "SKYDEW descent data are also valuable for identifying contamination." from the manuscript.

The current version of SKYDEW, with the sensing part located on the top of the flight package, is designed for ascent soundings. However, our experience with other soundings indicates that descent data is hardly affected by contamination. It's plausible that almost water that is adsorbed on the surfaces of the payload evaporates in the dry conditions of the stratosphere. Once the water has desorbed, the payload's surfaces do not accumulate contaminants in the dry stratosphere; this only happens when the instrument descends into the troposphere. If the water vapor and ice are completely removed from the payload, the SKYDEW can measure air free from contamination in regard to water vapor measurements. However, the evaporation process at the stratosphere can take some time due to the cold temperatures at those altitudes. Enough time is necessary to evaporate the contamination completely.

We cannot confirm whether the contamination has been completely removed at the stratosphere. Therefore, the sentence at line 628 will be deleted. It is necessary to consider a configuration for descent measurement in the future.

*Although the instrument is a frost-point hygrometer, could the authors please add mixing ratio*

*plots for the stratospheric sections of the profiles as they do in Figures 15 and 16? These panels could be added in almost all plots starting with Figure 7. In particular, the comparison plots in Figures 11-13 should show the mixing ratio and the relative mixing ratio difference for the stratospheric part of the profile (or a log plot for mixing ratio, which covers the entire altitude range).*

Yes, we will include the mixing ratio profile in Figures 11-13 in the revised manuscript. However, we will omit the mixing ratio in Figures 7-8, as these figures are primarily used to explain the golden point.

*Data availability: Thanks to the authors for making some data available from the YMC-BSM campaign. The GRUAN data archive does not make frost-point observations available and is not a suitable archive for that purpose. It would be better if these data were publicly archived elsewhere.*

We are currently in the process of developing the GRUAN data product for SKYDEW. However, since this development is not yet complete, these data have not been published on the GRUAN data server. Therefore, they are currently available only upon request. The soundings with RS41 include the XDATA packets in the RAW file. We plan to consider archiving the data used in this manuscript on an online data server.

*Figures:*

*Figure 1: The actual sensor is very hard to see. The right-hand panel could be enlarged to make the sensor arrangement clearer, in particular the coupling of the Peltier element to the large radiator. It appears as if there is a cover over the actual sensor. This cover may accumulate contamination that could potentially be detected by the mirror due to its proximity. This should be discussed.*

We will enlarge the right panel for better visibility. We will also add an explanation about the sensor cover in the manuscript. The revised manuscript will include the following sentence: "The sensor cover is made of aluminum and is painted black on the inside to prevent interference with the scattered light signal. The size of the cover is minimized to avoid the accumulation of water and ice."

*Figure 2 should show the first two panels with a wind speed of 5 m/s (typical balloon ascent rate), rather than 2 m/s or 0 m/s. These parameters will give a much better representation of the*

*temperature gradients that may be achieved in a sounding.*

Due to limitations in our experimental facility, we don't have data for 5 m/s at low temperatures and low pressure. The data for 5 m/s is available only at room temperature. For the experimental results, we will add a theoretical curve at -70°C and 5m/s, as explained earlier.

*Figure 4: The images of the condensate on the mirror do not come out well and could benefit from some contrast enhancements. Although the authors specify the air temperature at about 0°C, the mirror temperature is the relevant temperature (about -12°C). There is a red area underneath the enlargements, which also changes between the images. What does it represent?*

Thank you for your suggestion. We will adjust the contrast and brightness of the images for better clarity.
We will also add information about the mirror temperature in the manuscript as follows: "Figure 4 shows the behavior of the phase transition at an air temperature of approximately 0°C and a mirror temperature of -12 to -13°C."

[Figure]

Figure R1-7. Adjusted the images in the Figure 4.

*Figure 5: The caption lists the air temperature but should better give the mirror temperature at which these images were taken (they are listed in the legend of each panel). The text referring to this Figure should also list the mirror temperature, not the air temperature.*

We will include the mirror temperature data along with the air temperature in the revised manuscript.

*Technical comments:*

We will correct as your suggestions in the revised manuscript.

*Line 43: Replace "reduces" with "decreases".*

*Line 49: change to "… employ capacitive RH sensors."*

*Line 85: replace "cooler" with "colder".*

*Line 136: Replace "at" with "a".*

*Line 235: Delete "that".*

*Line 545: Insert:   … almost "always" …*

*Line 561: appear*

*Line 588: Figure 15 (not Figure 16)*

*Line 590: Replace "trail" by "train".*

*Line 615: Replace "over" with "of more".*

*Line 621: Remove the comma.*

*Line 624: "A dual sounding …"*

*Temperature differences are sometime expressed in °C and sometimes in K. Please use K for temperature differences throughout.*

*When referring to Figures, please use "Figure x" consistently and avoid using "Fig. x".*

---

## Author Comment (AC2)

**Reply to comments by Anonymous Referee #2**

The authors appreciate for your constructive comments and suggestions. We will revise the manuscript by taking your comments into account.

Below are the *reviewer #2 comments* in blue and our response in black.

*General Comments*

*The authors of the manuscript titled 'Development of a Peltier-based chilled-mirror hygrometer for tropospheric and lower stratospheric water vapor measurements' describe SKYDEW, a new chilled-mirror hygrometer for weather balloons that measures water vapor from the ground up to about 25 km altitude using thermo-electric cooling. The maximum altitude is constrained by the instrument's ability to dissipate heat and by outgassing from the balloon flight train. The latter is a common issue for balloon hygrometers.*

*The instrument and its principle of operation are documented comprehensively in the text and in the figures, together with a well-funded uncertainty analysis. The data processing is inspired from a 'golden point' approach introduced recently by a Swiss research group. The processing departs from the traditional averaging techniques and appears to be well suited for SKYDEW's aggressive PID controller, achieving frost point retrievals with a vertical resolution that I estimate from the figures about 50 m. It is unclear from the text how much of the processing is automatic and how much is based on user input. User input is not uncommon for this type of instruments.*

Regarding automatic processing, most of the processes are currently automated and do not require subjective analysis. However, the validation of the detection of the phase transition from water to ice is still insufficient. As explained below, we can distinguish the phase on the mirror by analyzing the behavior of the scattered light. Nevertheless, it still requires more comprehensive and diverse validation with the radiosonde relative humidity (RH) values under various conditions.

*SKYDEW is able to measure several kilometers into stratosphere, which is an improvement compared to Snow White, a thermo-electrically cooled balloon hygrometer of previous generation. The approach of using different operating set points for three different regions of the atmosphere is new, as far as I know, and is reasonable.*

Regarding the measurement limitations, it has been observed that SKYDEW can measure

altitudes over 25 km in many areas, excluding the polar regions at daytime. Under specific conditions, such as those in Lindenberg during nighttime, it has been demonstrated that measurements can be made up to nearly 30 km.

*SKYDEW's detector is located on top and outside of the instrument housing, similarly to the night version of Snow White. This design may allow for a better ventilation and heat dissipation and overall a faster response. On the other hand, the open design potentially increases the fragility of the sensor and may increase the risk of self-contamination when flown through saturated layers of the atmosphere, as the comparison with MLS retrievals shown in the text suggests. In addition, placing the sensor at the top of the instrument prioritizes ascent data over descent data.*

We decided not to incorporate an inlet tube, such as those used in CFH and NOAA FPH, in order to avoid the possibility of the tube itself becoming a source of contamination. We believe that the contamination indicated in the comparison with MLS retrievals is likely due to the balloon and string, rather than self-contamination.

SKYDEW was originally designed for ascent sounding; hence the sensing part is located at the top of the instrument package. For descent sounding, ideally, SKYDEW should be attached in an upside-down orientation, similar to the FLASH instrument. We are currently developing a special hanging system to switch the instrument-package direction for ascent and descent sounding.

*It is unfortunate that only one dual sounding is used for the reproducibility evaluation presented in this paper. The unexplained difference in the stratosphere of about - 0.5 K (or ~ 10%) w.r.t. to the CFH reference is also unfortunate, albeit based only on a very limited number of soundings and only one with CFH and SKYDEW on the same balloon. It appears that a full evaluation of the measurement accuracy of SKYDEW will be only possible further down the road, when more data is available. Nevertheless, the results presented in this initial paper are encouraging and the calculated uncertainties seem reasonable.*

We have conducted over 10 comparisons with the CFH so far (Sugidachi 2019). The profiles from the 5 soundings, excluding sounding data with some doubtful data in either of the two instruments, show that the SKYDEW readings are slightly smaller (less than approximately 0.5K) compared to the CFH readings in the lower stratosphere. However, these data have not yet been analyzed using the golden point method.

We will re-analyze these data using a new algorithm, which we will modify based on the reviewer's comments, and present the results in the revised manuscript.

References:

Sugidachi, T.:Meisei SKYDEW instrument: analysis of results from Lindenberg campaign, in 11th GRUAN Implementation and Coordination Meeting, Singapore, 23 May 2019 https://www.gruan.org/gruan/editor/documents/meetings/icm-11/pres/pres_0803_Sugidachi_SKYDEW.pdf (last access: 2 June 2024)

*The text is scientifically sound and within the scope of AMT, well structured, with some typing, grammar and referencing errors. While the presentation quality of some of the figures may be improved (especially in terms of dpi) they are readable and understandable.*

*The revision should correct technical errors and expand Section 3 with a few more clarifications.*

We will make sure to correct the typing, grammar, and referencing errors as noted in your specific comments in the revised manuscript. We appreciate your thorough review and attention to detail.

*In view of MLS decline and of the phase-down of hydrofluorocarbons, this presentation about an instrument capable of measuring lower stratospheric water vapor with thermo-electric cooling is relevant to the scientific community.*

*Specific comments*

*1: you may consider adding 'SKYDEW' into the title of the paper*

We will add 'SKYDEW' into the title as below.
"Development of a Peltier-based chilled-mirror hygrometer **SKYDEW** for tropospheric and lower stratospheric water vapor measurements"

*51: consider rewriting 'perform poorly in the dry stratosphere'. Operational radiosondes have been used in moist stratospheric conditions (e.g. Vömel et al. 2022).*

We will revise as suggested.

*67: 'these remote sensing techniques'. Which other ones? You have only mentioned Raman lidar in this paragraph.*

We will revise as suggested. In the revised manuscript, 'these remote sensing techniques' is

changed to 'this remote sensing technique'. Because the remote sensing sensors are generally calibrated using with the in-situ instrument such as radiosonde, we had used plural form "techniques" here.

*133 / Figure 1: Interesting design, with the sensor placed on top and outside the instrument, without inlet/outlet tubes. This departs from CFH/FPH. Could this explain the contaminated profiles shown in Figure 15, or is this due to contamination from the balloon and its flight train? On the other hand, the SKYDEW design may allow better ventilation and heat dissipation, and overall a faster response? Is this assumption correct?*

As mentioned in response to the general comment, we believe that the source of contamination indicated in Figure 15 is mainly from the balloon and string, not self-contamination. Most parts of the sensor, including the radiator and copper plate, except for the sensor cover, are thermally connected to the hot side of the Peltier device. This means that these parts are always warmer than the ambient air during ascent sounding, making it difficult for water to attach to the sensor parts.

However, the rig and string used to secure the SKYDEW and radiosonde may become contaminated in cloud conditions. Even with the configuration that places the SKYDEW sensor part at the top, the balloon and string can still be a source of contamination.

The design that places the sensor outside allows for better ventilation to release the heat from the hot side of the Peltier device. However, during the daytime, solar radiation heats the radiator, which can affect the lower cooling limit.

*What is the 'cover' in Figure 1 (c) and what is it made of? Is it used to guide the airflow over the detector? And/or to act as a protection against hydrometeors when flying in clouds?*

The sensor cover is made of an aluminum plate, and it serves two main purposes. First, it prevents direct solar radiation from entering the optical detector, as strong direct solar radiation can cause errors, even if the light source is modulated. Second, it is used to guide the airflow, as you mentioned.

We will add an explanation about the material and purpose of the sensor cover to the revised manuscript.

*The hot side radiator looks interesting. What is the purpose of the two 'screws' on the upper part of the radiator? For heat dissipation?*

The two screws on the upper part are for the wire to fix the radiator. For the previous version of SKYDEW, the ethanol was used for additional cooling of the radiator. This wire was used for the

guidance to ethanol tube. For the current version, this wire is used only for fix the radiator.

*136: Figure 2 suggests a maximum achievable cooling of about -50 C, not -90 C.*

We will revise as suggested.

*139: What is the wavelength of the light source and what is the modulation frequency? What is the technology (e.g. LED + Photodiode)?*

The optical sensor module in our system uses a red LED and a photodiode, with a peak wavelength of 660 nm. However, the modulation frequency is not publicly disclosed.

*140: What is the overall size (cm x cm x cm) of the SKYDEW instrument?*

The dimensions of the SKYDEW are 128 mm(W) x 93 (D) x 300 (H). We will add this information in the revised manuscript.

*146: The intensity of the scattered light is used in the processing and in the uncertainty analysis (Sections 3 & 4). Why do you consider it 'housekeeping data'?*

Because SKYDEW is an instrument designed to measure the dew/frost point, other parameters were considered auxiliary data. However, the intensity of the scattered light is particularly important for analyzing the dew/frost point profile, especially when compared with other housekeeping data. Accordingly, we will modify line 146 in the manuscript as follows:
  "Except for the mirror temperature and the intensity of the scattered light, these parameters are used as housekeeping data to monitor whether the system is working properly. The mirror temperature is directly linked to dew/frost point and the intensity of the scattered light is essential parameter to estimate dew/frost point correctly as described in Section 3 and 4."

*175: Incorrect, liquid water has a vapor pressure higher than ice, not lower.*

We will revise this sentence as suggested.
"Below 0 °C, liquid water has a vapor pressure higher than that of ice"

*204: Poor reference for specific humidity and precipitable water. In fact, this sentence may be removed entirely, as it provides no added value to the manuscript.*

We will delete this sentence as suggested.

*226: In Figure 2, do the dashed lines correspond to parametrizations using Equations 7 and 8? If so, please mention this and provide parameter values.*

No, the dashed lines are not theoretical values derived from Equations (7) and (8); they are simply fitting lines created using a quadratic function. As you have correctly pointed out, the meaning of these dashed lines is not clear. Therefore, we will add the theoretical lines derived from Equation (7) for greater clarity in the revised manuscript. Furthermore, the values at the lower stratosphere (-70°C, 100hPa, and 5m/s ascent) will be represented as a grey line in right panel. Estimating the heat transfer coefficient under no wind conditions is complex, so we have not indicated the theoretical line in the middle panel.

[Figure]

Figure R2-1. The revised figure about the cooling performance. The dashed lines indicate the theoretical values calculated by Eq. (7), using the parameter $\alpha = 0.0099, \beta = 0.0432, R_e = 0.48,$ $and\ S = 0.00025$. See the detailed information shown within each panel for different symbols, colors, and lines.

*288: Do you mean here that during a phase transition, you can distinguish if the mirror temperature corresponds to the dew point or to the frost point solely from the behavior of the scattered light?*

Yes, we can distinguish the phase on the mirror by observing the behavior of the scattered light. The automatic processing system uses the oscillation period of the scattered light, i.e., the interval between golden points. When the water on the mirror changes into large ice particles, the controller's oscillation period becomes longer, meaning that the intervals between Golden Points (GP) also become longer. The heating control creates fine ice particles, making the periods shorter again.

After launch, the point where the GP interval rapidly become longer is identified as the phase

transition timing. To be more specific, when the interval of the Golden point, as indicated by the gray line in Figure 9(d), becomes larger than a set threshold compared to the initial value, it is considered that a phase change has occurred. Tentatively, the threshold is set at 15 m. In many cases, the transition point corresponds to that estimated from the comparison with the radiosonde relative humidity (RH) value. For instance, in Figure 9(d), the interval of GP shows a sharp increase at approximately 3.5km, which indicates the phase transition.

*293: Is an intentional heating at -12.5°C also performed in NOAA FPH? Hall et al. 2016 do not mention it.*

This statement is indeed incorrect, thank you for pointing that out. Hall et al. (2016) do not mention it. They detect the phase transition by comparing with RH sensor and behavior of the mirror temperature. We will modify this sentence as follows.
"For the CFH , the forced freezing algorithm is applied at a mirror temperature of $-12.5$ °C to eliminate the phase ambiguity."

*321: From what I understand, Equation (11) fits to the 'golden points' in Section 3, i.e. the condensate on the mirror neither grows nor shrinks at the frost point. What is actually the relationship between the (mean) size of the ice crystals and the scattered light intensity in SKYDEW? Consider writing a few sentences why the 'golden points' of Poltera et al. 2021 apply to SKYDEW's scattered light signal.*

At the golden point, the left term of Equation (11) should be equal to zero, which results in $T_{fp}=T_m$. Therefore, the golden point can be applied to all chilled-mirror hygrometers in principle. We will add the following sentence at line 356: "Here, the golden point represents the equilibrium points when the condensate on the mirror neither grows nor shrinks, which corresponds to the evaporation rate in Equation (11) being zero."
Regarding the relationship between ice crystals and scattered light intensity, we can estimate the scattered light intensity from the ice particle, assuming backscattering by Mie scattering as described at line 301. By estimating the size and number of ice particles at each condition shown in Figure 5, we can say that many ice particles form on the mirror when the size is small to compensate for the weak signals from individual small ice particles.
As a rough estimation, at a temperature of -20 °C, the radius of the condensate is about 5-10 um and the number is 75 counts/mm². At a temperature of -40 °C, the radius of the condensate is about 3-5 um and the number is 400 counts/mm². At a temperature of -60 °C, the radius of the condensate is about 1.5-3 um and the number is 2880 counts/mm². These mirror conditions with different ice particles result in roughly the same order of total scattered light intensity. However, the speed varies according to Equation (11).

*331: ice 'crystals', not 'droplets'.*

We will revise as suggested.

*364: Is the removal of intentional heating data performed automatically by software?*

Yes, the removal of intentional heating data is performed automatically by the software. During the heating controls, the Peltier current is less than 0A, which indicates the heating of the mirror. Simultaneously, the scattered light signal drops below the target level (base level + 0.3 V for the first step and base level + 0.25 V for the second step) and approaches to the base level. If this behavior is detected, the mirror temperature data are removed for a duration of 40 seconds. Also, in case of reaching the cooling limit, where the Peltier current is at its maximum (>2.3 A), and the scattered light drops below the base level +0.15 V, the mirror temperature data is also removed.

*383: This paragraph needs further clarification. Do you extract the 'golden points' on the smoothed mirror temperature, or from the original mirror temperature? From Figure 7 (upper left), it seems that the points are extracted from the original profile, but this is not how I understand the text. Moreover, how do you perform the final smoothing of the extracted 'golden points'? They seem to come at irregular time intervals. Do you linearly interpolate between the 'golden points' on a 5 Hz time axis and then smooth with a Gaussian filter of sigma=1.5 seconds?*

A Gaussian filter was applied to the original data even though they are with irregular time intervals. We will modify the algorithm, including this smoothing process, in the revised manuscript according to the reviewer's comment. The main modifications are as follows: (1) Up sampling is not applied; the golden point is detected from 1Hz sampling data. (2) The raw selected golden points are resampled at regular 1 Hz points with interpolation.

We have also realized that the caption is incorrect - this example profile is not the same as the one in Figure 6. The profile in Figure 7 was the sounding result on 5 November 2019. We will correct this and include a more detailed explanation in this section in the revised manuscript.

[Figure]

Figure R2-2 As for the Figure 7 in the original manuscript, but with the addition of black dots indicating the detected raw golden points.

*430: In Table 1, you write ' ~ 1.0 K at max' for this uncertainty component, why not write this here? Moreover, what about the typical uncertainty value when oscillations are 'normal'?*

We will modify the value in Table 1. We will add the typical two values for the normal case and for the large oscillation case separately. But, these values will be changed in the future according to the modification of the algorithm.

**Table 1:** Uncertainty sources and estimates for SKYDEW.

| Uncertainty parameter | Value (k=1) | (Un)/correlated |
|---|---|---|
| Mirror_temp measurement, u_mirror | $0.052=(u\_mrr1^2+u\_mrr2^2+u\_mrr3^2)^{0.5}$ | correlated |
|    PT temperature calibration, u_mrr1 | $0.07/\sqrt{3}$ | correlated |
|    Resistance measurement u_mrr2 | $0.005\sqrt{3}$ | correlated |
|    Thermal gradient of mirror, u_mrr3 | $0.015/\sqrt{3}$ | correlated |
| Golden point detection error, u_GP_error | **> 1.0K in case of the large oscillation**
**0.2K at normal** | uncorrelated |
| Filtering deviation, u_filter | **>0.5K in case of the large oscillation**
**<0.1K at normal** | uncorrelated |

| | | |
|---|---|---|
| Contamination by cloud | Depend on the situation strongly. | correlated |
| Aerosol effect and curvature effect | negligible | correlated |
| Total uncertainty, u_DP | $=(u\_mirror^2+u\_GP\_error^2+u\_filter^2)^{0.5}$ | |

*452: Intentionally heating the mirror first at -12.5 C, such as CFH, seems reasonable for an ascending balloon, as it almost certainly ensures that the mirror condensate is liquid water before the heating stage and ice after it. SKYDEW heats its mirror first at -36 C, i.e. at a temperature where the condensate has very likely already phase-transitioned to ice, which complicates the mirror condensate phase determination. Why not perform the first heating stage at a lower temperature in SKYDEW?*

The reason we chose to initiate the intentional heating at -36°C with SKYDEW, despite it being a temperature where the condensate has likely already phase-transitioned to ice, is due to our past experimental results. When we attempted to initiate the intentional heating at -12.5°C, as done for CFH, we often experienced failures where ice particles did not form, and super cooled water droplets remained on the mirror for an extended period.

Therefore, we selected a temperature where we could be certain that ice particles would form. As mentioned earlier, the phase transition can be detected automatically. Our intention was to avoid having large ice particles, formed from water during the phase transition, remaining on the mirror for extended periods. Such large ice particles are not suitable for the PID controller in SKYDEW.

*From this paragraph, it is unclear how you determine the phase of the condensate in practice. From the amplitude and frequency of the scattered light fluctuations? Or from the comparison with the partnering radiosonde RH sensor? Or both? Can this be performed automatically by software?*

Yes, we can distinguish the phase on the mirror by monitoring the behavior of the scattered light. The automatic processing system uses the oscillation period of the scattered light, i.e., the interval between golden points. When the water on the mirror changes into large ice particles, the periods driven by the PID controller become longer, meaning the intervals between Golden Points (GP) also become longer. The heating control creates fine ice particles, causing the periods to shorten again.

The point where the GP interval rapidly lengthens is considered as the phase transition timing. Currently, the threshold is set at 15m. In many cases, the transition point corresponds to that estimated from the comparison with the radiosonde relative humidity (RH) value. For instance, in Figure 9(d), the interval of GP shows a sharp increase at approximately 3.5km, which

indicates the phase transition.

455: Incorrect, cubic ice on the mirror would cause a lower mirror temperature.

We will revise as suggested.

504: How large is the air temperature uncertainty uT? Please provide a typical value or an upper bound

The air temperature uncertainty for this particular sounding is approximately 0.2 K. The uncertainty tends to increase at higher altitudes during the daytime due to the influence of solar radiation. However, at nighttime, the uncertainty remains fairly constant from the surface to the stratosphere (as per Kizu et al. 2019). Most SKYDEW soundings, including the sounding mentioned in your question, were performed at nighttime. We will add the typical value in the revised manuscript.

511 / Figure 9 (h): the uncertainty due to pressure is surprisingly small compared to e.g., Hall et al. 2016 for FPH? Why is that? How large is the air pressure uncertainty uP? Please provide a typical value or an upper bound

For this particular sounding with the RS-11G radiosonde, the pressure uncertainty is estimated to be less than 0.15 hPa in the stratosphere. The pressure of the RS-11G is derived from the GPS height. On the other hand, the iMet-1-RSB and RS-80, which are used for the NOAA FPH, are equipped with a pressure sensor. The uncertainties of these devices seem to be larger than that of the RS-11G, especially in the stratosphere. In the revised manuscript, we will add the typical value for the pressure uncertainty in the stratosphere.

567: You mention at the beginning that several SKYDEW-CFH comparisons have been performed, yet only two soundings are discussed, only one on the same balloon. Is the -0.5 K difference in the stratosphere systematic in all soundings so far, or does it appear only on these two soundings?

As mentioned above, we will add the comparison with CFH in the revised manuscript.

588: the MLS data has been averaged over how many +- hours or days?

In the original study, all data from the same day as the balloon release were used. However, in the revised manuscript, we will modify the comparison results with the MLS data using a more appropriate averaging method. This will be reanalyzed by referring to Hurst et al. (2023), which uses temporal and spatial coincidence criteria of +/-18 hours, +/-2° latitude, and +/-8° longitude. Additionally, we will use the same average kernel vertically as the MLS data.

*596: The measurements took place during the summer monsoon, which probably increases the risk of contamination for this type of instruments. Nevertheless, do you know where the outgassing that you mention here comes from? Is it the SKYDEW sensor probe itself that suffered from icing in the saturated troposphere? Or is it from the balloon and its flight train? > 50 m train lines are generally recommended for measuring stratospheric water vapor, have you used such longer lines?*

We cannot definitively determine the source of the contamination from outgassing in this case. Given that the balloon and string are positioned above the sensor part of SKYDEW, outgassing from these parts could easily be a source of contamination. The radiator is close to the mirror part, but it is made of metal and is warmer than the air temperature. Therefore, it is not easy for water vapor to be attached onto this surface. However, water droplets and ice particles in clouds may attach to the sensor part. It is recommended to avoid cloudy conditions for stratospheric water vapor measurements, in addition to rainy conditions.

We usually use a 50m unwinder for most soundings, including those at the YMC-BSM campaign. However, in certain limited cases, such as when the landing point is estimated to be near a city area in Japan, we use a 30m unwinder and a 600g balloon for safety reasons.

We will revise line 439 accordingly.

*You mention that the soundings on 1 & 3 June passed through saturated layers, but what happened on 27 May? What is the reason for the disagreement with MLS above 20 km on that day? The 27 May sounding has for example not been excluded from the comparison with RS11G on Figure 14.*

There is a high possibility that the payload passed through a thin cloud layer during the sounding on 27 May as well as 1 & 3 June. The oscillation in the scattered light signal at approximately 13 km corresponds to the near-saturated layer shown in the frost point profile. When SKYDEW passes through a cloud layer, such oscillation often occurs, especially in the upper troposphere, due to the aggressive PID tuning. The sounding on June 1 was similar to that on June 3, with a thick cloud layer present in the upper troposphere.

Contamination from clouds in the upper troposphere does not significantly impact the measurements in the troposphere. More specifically, while such contamination may cause a

slight bias above the cloud layer, we have included the results from these soundings in Figure 14. This is because the effect on the comparison at the troposphere level is considered minimal.

We will modify lines 594-596 as follows: "For the soundings on 27 May, 1 June, and 3 June, the SKYDEWs passed through saturated air or cloud layers in the upper troposphere. There is a possibility that the ascent measurements of these two profiles were affected by contamination from outgassing."

[Figure]

Figure R2-2 the example of the contaminated soundings

*629: Why is controlled descent a challenge for SKYDEW? Is this because SKYDEW might lose its condensate before balloon turnover? Or is it because of the position and orientation of the sensor?*

There are indeed two main challenges as you mentioned. Firstly, it takes a long time to form condensate on the mirror at stratospheric altitudes due to the limited cooling capacity of the Peltier cooler. Therefore, the balloon's turnover needs to occur just before the loss of the condensate, and this altitude can vary depending on atmospheric conditions. An algorithm that allows SKYDEW to detect the cooling limit and control the balloon's ascent/descent is necessary. Alternatively, the balloon should be turned over at a sufficiently low altitude before the condensate is lost.

In addition to ensuring that the condensate is not lost, the sensor's orientation should be upside down for descent soundings. This issue is common for operational radiosondes such as the RS-11G and RS41, as most of them are designed for ascent soundings. A system to switch the orientation of the payload is needed for accurate measurements during descent.

We will revise the last sentence to: "because the current version of SKYDEW is designed for ascent measurements only."

*Technical corrections*

*Here a list of typing and referencing errors that I was able to spot, sorted by line number or figure number.*

We will correct these typing and referencing errors in the revised manuscript as per your comments. Thank you for pointing them out.

*21: selects the equilibrium point*

*25: in regions*

*46: The Brewer-Dobson circulation*

*63: integration time*

*66: Whiteman et al. 2006 is missing from the reference list*

*89: Hurst et al. 2023*

*90: I have found the NOAA instrument description in Vömel et al. 1995 (10.1029/95JD01000), not Vömel et al. 1995 (doi:10.1029/95GL02940).*

*93: water vapor concentrations. Generally, stick to either American or British.*

*96: cooperation agreement*

*98: the Fluorescent Advanced Stratospheric Hygrometer*

*98: the Vaisala RS92 radiosonde*

*110: analog*

*123: Put the 3 in CFH3 in subscript*

*129: Section 5*

*129: Section 6*

*136: a temperature difference*

*150: Vaisala RS41, Intermet 54*

*162: You have two Vömel et al. 2016 references, please distinguish between the two in the text*

*172: You have three Vömel et al. 2017 references, please distinguish between the three in the text*

*174: Fujiwara et al., 2003*

*181: Vömel et al. 2007 mention -53 C, not -55 C.*

*182: You have three Vömel et al. 2017 references, please distinguish between the three in the text*

*193: WMO (2021)*

*216: between the mirror surface and ambient air, and S*

*220: thermal conductivity of air,*

*225: For higher air temperature*

*230: high cooling power is needed*

*235: (PT100) which has*

*251: the control output corresponds to the current*

*259: A proportional controller alone cannot eliminate*

*266: which depends*

*276: at an air temperature of ~0 °C and at ~-13 °C dew point.*

*285: This implies that the output of the PID controller*

*308: As in the cloud formation process, the number of IN on the mirror may reflect the temperature dependence of the ice-nucleation process on the mirror.*

*334: at an as low as possible*

*343: when the mirror temperature reaches about −36 °C.*

*406: remove underscore before 'to reduce this distribution'*

*462: (Thornberry et al., 2011).*

*467: Thornberry et al (2011)*

*472: Pruppacher and Klett, 1997*

*484: Kizu et al. (2018) is missing from the reference list*

*492: in long time series*

*540: A dual sounding with two SKYDEWS*

*543: worked properly.*

548: Smaller oscillations are required for better measurements, although the oscillations due to the aggressive setting of the PID controller are needed to detect the golden points.

559: The deviation is large in the troposphere

611: dynamic changes of atmospheric water vapor.

615: The Peltier cooling creates a temperature difference of more than 40 K

726: Missing new line between '2022' and 'Sakata, R.'

744: Vömel

794: which corresponds to the right axis.

796: Photographs show the condensate on the mirror.

799: Condensates on the mirror at air temperatures of

819: the detected golden points

831: Profile of dew/frost point

837: Frost point profile on 26 November 2021

863: The center panel shows

875: Gray shading indicates the uncertainty of SKYDEW (k = 2).

877: Mirror temperature measurement, u_mirror

878: Resistance measurement u_mrr2: missing 'divide' sign between 0.005 and ¥sqrt{3}

Figures 6 (b) and 9 (b): The sign of the Peltier current is inconsistent w.r.t Figure 2.

Figure 14 (a): The 'black dashed line' appears to be missing in Figure 14 (a). Please add the line in the figure, or remove this sentence from the figure caption.